# Antimicrobial peptide expression in a wild tobacco plant reveals the limits of host-microbe-manipulations in the field

Arne Weinhold[1]*[†], Elham Karimi Dorcheh[1], Ran Li[1], Natarajan Rameshkumar[1,2], Ian T Baldwin[1]

[1]Department of Molecular Ecology, Max Planck Institute for Chemical Ecology, Jena, Germany; [2]Biotechnology Department, National Institute for Interdisciplinary Science and Technology, Thiruvananthapuram, India

**Abstract** Plant-microbe associations are thought to be beneficial for plant growth and resistance against biotic or abiotic stresses, but for natural ecosystems, the ecological analysis of microbiome function remains in its infancy. We used transformed wild tobacco plants (*Nicotiana attenuata*) which constitutively express an antimicrobial peptide (Mc-AMP1) of the common ice plant, to establish an ecological tool for plant-microbe studies in the field. Transgenic plants showed in planta activity against plant-beneficial bacteria and were phenotyped within the plants´ natural habitat regarding growth, fitness and the resistance against herbivores. Multiple field experiments, conducted over 3 years, indicated no differences compared to isogenic controls. Pyrosequencing analysis of the root-associated microbial communities showed no major alterations but marginal effects at the genus level. Experimental infiltrations revealed a high heterogeneity in peptide tolerance among native isolates and suggests that the diversity of natural microbial communities can be a major obstacle for microbiome manipulations in nature.

DOI: https://doi.org/10.7554/eLife.28715.001

*For correspondence: arne.
weinhold@fu-berlin.de (AW)

**Present address:** [†]Applied Zoology/Animal Ecology, Dahlem Centre of Plant Sciences, Freie Universität Berlin, Berlin, Germany

## Introduction

Plants are surrounded by a vast and diverse community of soil bacteria, some of which are able to form close associations and important mutualistic relationships with plants (*Hardoim et al., 2015*; *Müller et al., 2016*). Plant-microbe interactions play an important role in plant health and productivity and have received increasing attention for their roles in natural ecosystems as well as in agriculture for their utilization in advanced plant breeding (*Busby et al., 2017*; *Hacquard et al., 2017*; *Kroll et al., 2017*). Many bacteria are considered to be either harmless or to benefit a plant under certain conditions, and some are suspected to be even involved in aboveground defenses against herbivores (*Badri et al., 2013*; *Humphrey et al., 2014*; *Schädler and Ballhorn, 2016*) or flowering phenology (*Wagner et al., 2014*).

However, most microbiota inhabit plants without producing symptoms, and despite the assumption of evolutionary benefits of the plant's holobiont, little is known about the ecological relevance of most plant-associated bacteria (*Müller et al., 2016*; *Sánchez-Cañizares et al., 2017*). Functional characterizations are usually limited to culturable bacteria, frequently used in gnotobiotic conditions or inoculated in titers higher than those of native soils and likely overestimating their real roles in nature (*Haney et al., 2015*). The reconstruction and establishment of artificial communities or microbial consortia refines this approach, but remains restricted to culturable bacteria (*Vorholt et al., 2017*).

Modern sequencing techniques, such as 454 pyrosequencing, enable a more comprehensive and culture-independent characterization of plant-associated bacteria and allow the in situ identification

**eLife digest**   Plants never grow alone in their natural environments, but are instead constantly surrounded by microbial life. The microbes associated with a plant are known as the plant's microbiome, and while a few of them are thought to benefit the plant's health, little is known about what most of these microbes really do. It is also not clear under which conditions so-called beneficial microbes would benefit a plant. This is partly because scientists do not know how a plant would look if it were missing these microbes in nature, as it has been impossible to grow a "microbe-free" plant under field conditions.

Now, Weinhold et al. have tested a new approach to manipulate a plant's microbiome in its natural environment. First, wild tobacco plants (*Nicotiana attenuata*) were genetically modified to produce an antibiotic substance, specifically an antimicrobial peptide. Instead of providing disease resistance to disease-causing bacteria, this peptide mainly inhibits the growth of bacteria that benefit plants. Then, these modified plants were grown and studied in glasshouses and in field experiments in southwestern Utah.

Although experiments in the glasshouse confirmed that the modified plants reduced the growth of certain beneficial bacteria, this was not equal among all strains tested. Even closely related bacteria belonging to a genus called *Bacillus* responded very differently. However, the field trials showed no negative consequences for the modified plants; they still grew to the same size as normal in their natural environment. Sequencing the genetic material of the microbiomes from those field-grown plants revealed that the bacterial community associated with the roots was not altered in a major way, but showed only subtle differences.

Together these findings show that, contrary to expectations, the attempt to manipulate a plant's microbiome in a natural environment had little impact on the plant and its microbiome. Weinhold et al. suggest that the rich diversity of bacteria in the soil may account for resilience of microbiomes in natural environments. Nevertheless, Weinhold et al. hope that their unusual approach can inspire other researchers to consider more innovative ways to study plant-microbe interactions in the wild. Also, these new findings still have direct implications for agriculture, because they alleviate long-held concerns that using antimicrobial peptides to protect crops might harm beneficial microbes and negatively affect plant growth.

DOI: https://doi.org/10.7554/eLife.28715.026

of previously overlooked communities (*Lundberg et al., 2013*). High-throughput sequencing technologies have revealed rare taxa and whole community compositions, and have greatly increased our understanding of microbiome assemblages in Arabidopsis, maize and rice (*Bulgarelli et al., 2012*; *Lundberg et al., 2012*; *Peiffer et al., 2013*; *Edwards et al., 2015*), but also for non-model plants within their native habitats (*Coleman-Derr et al., 2016*; *Fonseca-García et al., 2016*; *Wagner et al., 2016*). Soil type and geographical locations seem to be major determinants of microbiome variations, whereas plant cultivars or genotypes have a much smaller influence (*Peiffer et al., 2013*; *Edwards et al., 2015*). Plant-microbe interactions are complex, and we are just beginning to understand the factors which shape microbial associations and which are essential for bacteria to inhabit the intercellular space of a host plant (*Ofek-Lalzar et al., 2014*; *Levy et al., 2018*). Still, the ecological analysis of root microbiome function remains in its infancies (*Naylor et al., 2017*; *Fitzpatrick et al., 2018*), and the influences of even highly abundant (but unculturable) taxa of the plant microbiome remain unknown. Novel approaches are needed to provide experimental procedures which could link changes in community composition to fitness consequences under native growth conditions, to be able to utilize this knowledge for sustainable and targeted microbiome engineering (*Foo et al., 2017*; *Oyserman et al., 2018*). A 'microbe-free plant', not only as a theoretical game of thought, would be a valuable ecological tool to reveal hidden phenotypes of bacterial mutualisms under natural environmental conditions (*Partida-Martínez and Heil, 2011*; *Gilbert and Neufeld, 2014*). However, the feasibility of engineering a near-aposymbiotic plant using a transgenic approach, remains to be evaluated.

Antimicrobial peptides (AMPs) are small, cationic peptides, which have been shown to inhibit the growth of a broad range of microbes. They can be found among many organisms and are part of

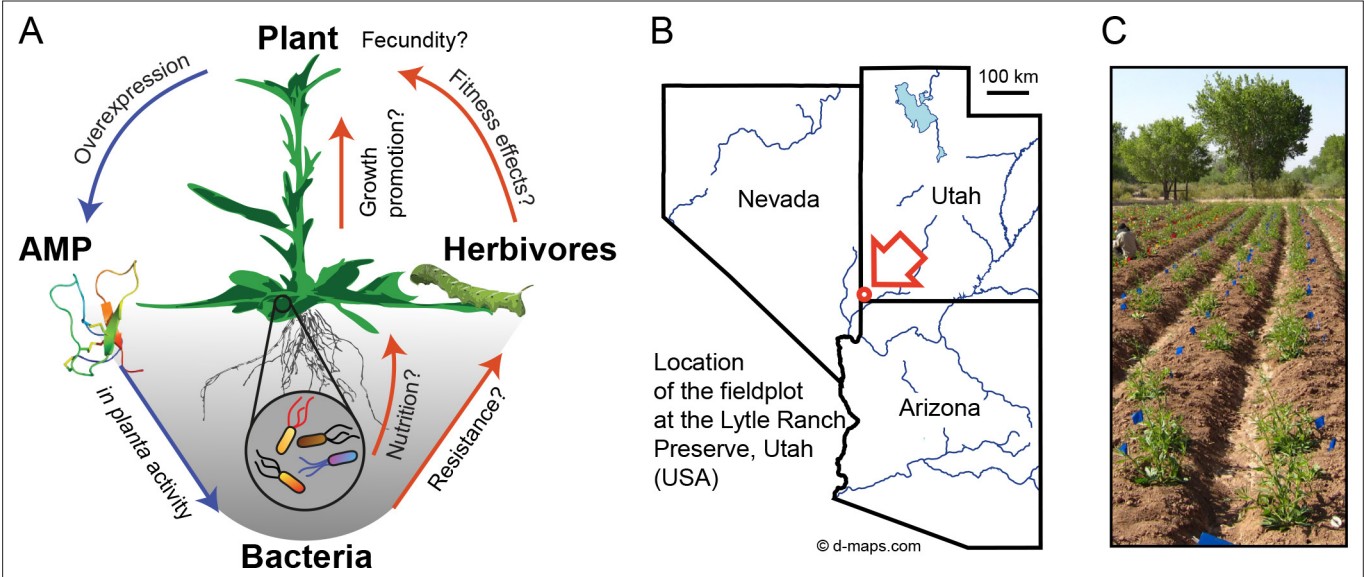

**Figure 1.** *Nicotiana attenuata* plants were transformed to ectopically express an antimicrobial peptide (AMP) and planted into a research field plot in the plants´ native environment. (**A**) Overview scheme of the experimental approach. AMP expression was used to target bacteria in planta as an ecological tool to unravel microbiome function under field conditions regarding contribution to plant performance, fitness and defense against herbivores. (**B**) Location of the experimental field plot at the Lytle Ranch Preserve in Utah (USA), which was used for the field releases in 2011, 2012 and 2013. Figure composed with maps provided by d-maps.com (http://d-maps.com/m/america/usa/utah/utah/utah09.pdf; http://d-maps.com/m/america/usa/arizona/arizona/arizona13.pdf; http://d-maps.com/m/america/usa/nevada/nevada/nevada09.pdf). (**C**) Larger view of the plants planted at the field plot.

DOI: https://doi.org/10.7554/eLife.28715.002

The following figure supplement is available for figure 1:

**Figure supplement 1.** Antimicrobial peptide (AMP) selection scheme for the identification of eligible candidates for host-microbe manipulation studies in the field.

DOI: https://doi.org/10.7554/eLife.28715.003

the innate immune system of plants as well as animals. This allows the expression of AMPs across kingdom levels, and raised expectations for their medical application or biotechnological engineering of pathogen resistance in transgenic plants (*de Souza Cândido et al., 2014*; *Ageitos et al., 2017*). For agricultural purposes, such approach has been employed for various plants, including tobacco, potato or Arabidopsis (reviewed in *Holaskova et al., 2015*). Little is known about the consequences that AMP expression has on beneficial plant mutualists, and the investigation of unintended effects is only rarely considered (*Meyer et al., 2013*).

As a novel approach to evaluate microbiome function under native growth conditions, we expressed an AMP with selective activity against a mainly beneficial subset of bacteria and investigated if this would influence plant performance in the field (*Figure 1A*). For this purpose, we expressed more than ten different AMPs from various origins in the wild tobacco plant *Nicotiana attenuata* (Torr. ex S. Watson). All transgenic plants were thoroughly screened to ensure reliable gene expression levels over multiple generations and the accumulation of the heterologously expressed peptides was confirmed using a novel nanoUPLC-MS$^E$ based quantification method as shown in previous publications (*Gase et al., 2011*; *Weinhold et al., 2013*, *Weinhold et al., 2015*) (*Figure 1—figure supplement 1*). The antimicrobial peptide 1 (Mc-AMP1) of the common ice plant (*Mesembryanthemum crystallinum*) proved to be an interesting candidate for this endeavor, as this peptide showed not only high accumulation levels within the leaf apoplast (*Weinhold et al., 2015*), but also antibacterial activity in planta, as demonstrated by experimental infiltrations within this study. The expressed peptide has an average molecular mass of 4.2 kDa and belongs, due to the distinct 'knot' connection motif of the disulfide bridges, to the knottin sub-family of antimicrobial peptides (*Sampedro and Valdivia, 2014*). These types of peptides are reported to have selective activity against gram-positive, but not gram-negative bacteria (*Pelegrini et al., 2011*). The PhytAMP database entry for

Mc-AMP1 (ID:PHYT00272) reports under in vitro conditions, activity against *Bacillus megaterium* and *Micrococcus luteus* (formerly *Sarcina lutea*), but no activity against *Erwinia carotovora* or *Escherichia coli*. Such a selective activity spectrum against *Actinobacteria* and *Firmicutes* would provide the opportunity to preferentially target a non-pathogenic subset of plant-associated bacteria, since the majority of the known plant pathogens are gram-negative (e.g. *Pseudomonas*, *Ralstonia*, *Agrobacterium*, *Xanthomonas*, *Erwinia*, *Xylella*, *Dickeya* and *Pectobacterium*), with notable exceptions among the *Actinobacteria* (e.g. *Clavibacter* spp.) and even a few *Bacillus* spp. (*Mansfield et al., 2012*). In contrast, the majority of *Bacillus* spp. described as plant endophytes or rhizosphere members, have been characterized as plant-beneficial bacteria, with disease suppressive or plant growth promoting abilities (*Gutierrez-Manero et al., 2001*; *López-Bucio et al., 2007*; *Hardoim et al., 2015*), whereas the colonization with certain *Actinobacteria* is believed to contribute to drought tolerance of some plants (*Naylor et al., 2017*; *Fitzpatrick et al., 2018*). Previous experiments have shown that the culturable bacterial community of *N. attenuata* is dominated by *Bacillus* spp. (*Long et al., 2010*), of which some had the ability to restore normal growth of ethylene-insensitive plants in the field, in part by supplying plants with reduced sulfur (*Meldau et al., 2012*, *Meldau et al., 2013*).

*N. attenuata* is a post-fire annual plant inhabiting the Great Basin Desert of the western USA, and has been used for decades as an ecological model plant for determining fitness effects of single genes within the plant's native habitat (*Steppuhn et al., 2004*; *Schuman et al., 2012*). Targeted genetic manipulation allows for reverse genetic studies to analyze the consequences of different defense strategies against biological or environmental stresses. As the natural history of *N. attenuata* has been intensively studied, this plant is an excellent system to explore ecological consequences of AMP expression using in-depth field studies in the native environment

Here we evaluate with an unbiased approach, if the ectopic expression of an AMP, which targets gram-positive bacteria, has consequences for the fitness of a wild plant regarding growth, herbivore infestation, reproductive capacity or the composition of the root-associated bacterial communities under field conditions (*Figure 1A*). A long standing concern for the use of broad-spectrum antimicrobial peptides in transgenic crops is that this could influence plant symbionts or other non-target groups of the soil microflora (*Glandorf et al., 1997*). However, these assumptions are usually based on AMP activity estimations from in vitro studies and empirical evidence from comprehensive field studies is still missing. We observed no major shift in the root-associated bacterial community and could demonstrate with experimental infiltrations that native bacterial isolates show distinctive and highly strain-specific susceptibilities, and concerns of broad-spectrum antimicrobial effects within transgenic plants are unfounded.

## Results

### Antimicrobial peptide expression in Nicotiana attenuata

We transformed *N. attenuata* plants for the ectopic expression of antimicrobial peptides, to evaluate potential effects on plant phenotype in nature (*Figure 1*). The antimicrobial peptide 1 (Mc-AMP1) from the common ice plant (*M. crystallinum*), was cloned behind a constitutive 35S promoter as previously described (*Gase et al., 2011*). This peptide consists of a signal peptide and a 37-amino acid long mature sequence with a knottin-like folding motif stabilized by three cysteine disulfide bonds (*Figure 2A*). Knottin-like peptides are known from plants, as well as animals and fungi and show high sequence similarities across these kingdoms (*Figure 2—figure supplement 1*) (*Daly and Craik, 2011*). The Mc-AMP1 expressing 'ICE' plants were thoroughly screened and plants from three independent transformation events selected (ICE 1, ICE 6 and ICE 8), which showed no evidence for transgene silencing (*Weinhold et al., 2013*). These lines were propagated as outlined in *Figure 2—figure supplement 2* to obtain single insertion lines from each transformation event, which were used for all following experiments. Single insertion lines retained uniform transgene expression in the $T_4$ generation and showed a coefficient of variation (CV) below 2% in rosette-stage leaves (*Figure 2B*). As demonstrated in a previous study, the heterologously expressed peptide accumulated in the leaf apoplast with average levels of 160 (±40) pmol × g fresh leaf mass$^{-1}$ (*Weinhold et al., 2015*). Immunoblot assays demonstrated peptide accumulation in the leaves as well as the roots of the transgenic plants (*Figure 2—figure supplement 3B*). All plants flowered and produced fertile seeds and the peptide accumulation in shoot and root tissue had no influence on plant growth and development

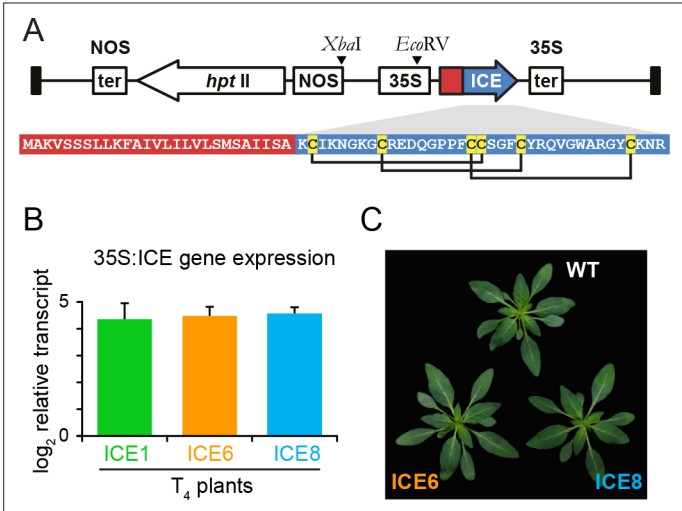

**Figure 2.** Overview of the expression casette used for plant transformation and the gene expression strength of the transformed *Nicotiana attenuata* plants. (**A**) The peptide Mc-AMP1 ('ICE') from the common ice plant (*Mesembryanthemum crystallinum*) was constitutively expressed under a 35S promoter. The amino acid sequence of the pro-peptide is shown with the signal peptide in red and the mature domain in blue. Connection pattern of the conserved cysteine residues are indicated. (**B**) Uniformity in gene expression strength in rosette leaves of independently transformed *N. attenuata* ICE lines in the $T_4$ generation. Bars indicate the $\log_2$ fold AMP expression levels which were 23.1 fold (±6.0) higher relative to actin as the reference gene (±SD, n = 4 plants). (**C**) The ICE lines showed no pleiotropic effects on plant morphology associated with AMP expression.
DOI: https://doi.org/10.7554/eLife.28715.004

The following figure supplements are available for figure 2:

**Figure supplement 1.** Multiple sequence alignments of knottin-like peptides.

DOI: https://doi.org/10.7554/eLife.28715.005

**Figure supplement 2.** Selection scheme for ICE overexpression lines harboring a single copy of the transgene.

DOI: https://doi.org/10.7554/eLife.28715.006

**Figure supplement 3.** Transgene expression and peptide accumulation had no pleiotropic effects on plant growth and development in the glasshouse.

DOI: https://doi.org/10.7554/eLife.28715.007

(*Figure 2C*) (*Figure 2—figure supplement 3A–C*). These ICE-overexpression lines were chosen for this study, since experimental infiltrations showed in planta activity against gram-positive bacteria.

## In planta activity against Bacillus, but not Pseudomonas

To evaluate the antibacterial activity of the transgenic plants in planta, we modified a leaf pressure infiltration/reisolation method to apply this for various different bacterial taxa (*Figure 3—figure supplement 1A*). Infiltrations with the gram-positive bacterium *Bacillus pumilus* DSM 1794 resulted in consistent pattern of colony forming units (CFUs) and was used for the standardized screening and selection of transgenic plants during glasshouse propagation. Only plants of the ICE-genotype showed clear reductions in CFUs from 3 to 9 days *post* infiltration (dpi) (*Figure 3—figure supplement 1B*), which allowed us to identify and eliminate plants that carry non-functional T-DNA insertions, as shown for line ICE 1.5.2 (*Figure 3—figure supplement 1C,D* red arrow). The three remaining functional ICE-lines with antibacterial activity (ICE 1.1.1, ICE 6.4.2 and ICE 8.4.1, hereafter simply called ICE 1, ICE 6 and ICE 8) retained the antibacterial phenotype within the $T_4$ generation with consistent CFU reductions of *Bacillus pumilus* from 2 to 6 dpi (p<0.0003, t-test, 6 dpi) (*Figure 3*). As a gram-negative strain, we used *Pseudomonas syringae* pv *tomato* DC3000 within the same assay. Here, we observed for the ICE-lines no difference in CFU counts and no increase in resistance of the transgenic plants compared to the controls (*Figure 3*). This observation was consistent with the expectation that

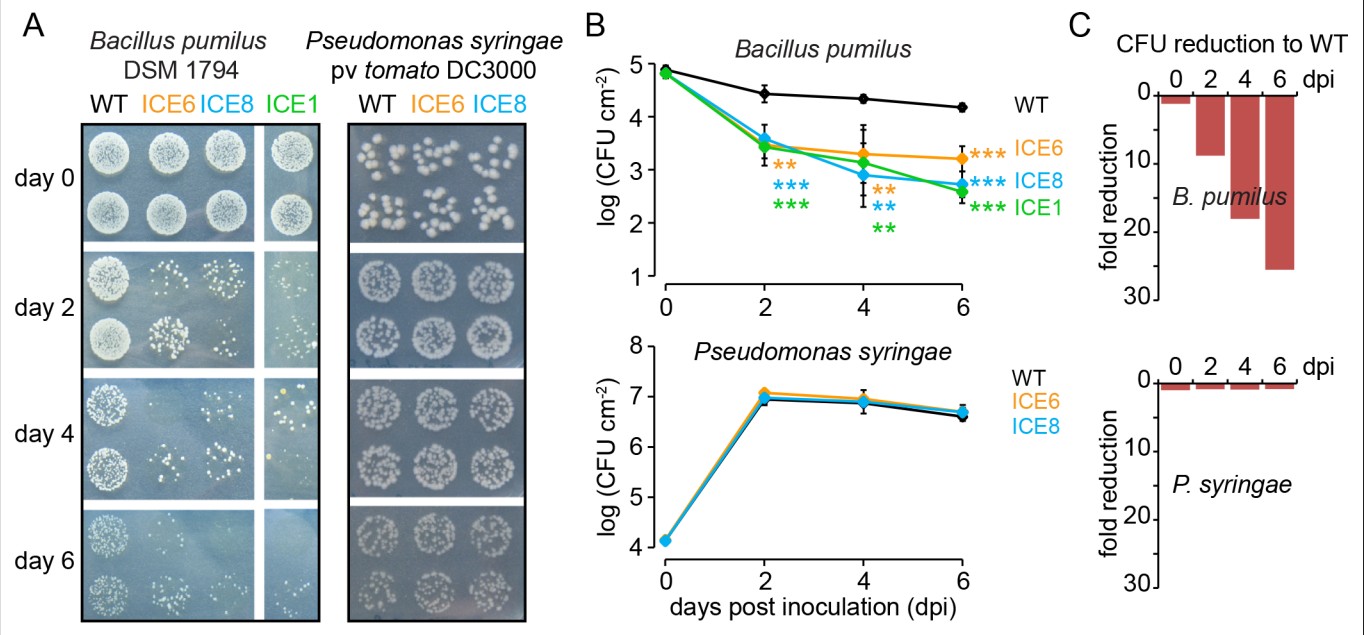

**Figure 3.** Transgenic *N. attenuata* plants showed in planta antimicrobial activity against *Bacillus pumilus* DSM 1794, but not against *Pseudomonas syringae* pv *tomato* DC3000. (**A**) Bacteria were pressure infiltrated into fully expanded rosette-stage leaves and re-isolated at 0, 2, 4 and 6 d *post* infiltration (dpi). The pictures show two technical replicates per genotype from a dilution series. (**B**) The mean colony forming units (CFU) were plotted as log CFU cm$^{-2}$ leaf area (±SD, n = 4 plants). Asterisks indicate statistically significant differences between WT and the transgenic plants (students t-test; **p≤0.01; ***p≤0.001). (**C**) Summary of the CFU fold reduction (non-log scale) compared to WT averaged among genotypes.
DOI: https://doi.org/10.7554/eLife.28715.008

The following figure supplement is available for figure 3:

**Figure supplement 1.** Leaf infiltration method for the determination of in planta antimicrobial activity against *Bacillus pumilus* DSM 1794.

DOI: https://doi.org/10.7554/eLife.28715.009

the expression of the 'ICE' peptide results in selective antimicrobial activity against gram-positive, but not against gram-negative bacteria.

## AMP expression has no effect on plant growth nor canopy damage by native herbivores in the field

We wanted to evaluate if the expression of this peptide could influence plant-beneficial bacteria, which could have unexpected ecological consequences when the plants are grown in their native environment. During the first field season we planted two independent lines of this genotype (ICE 1 and ICE 6) together with empty vector controls (EV) on a research field plot located at the Lytle Ranch Preserve in Utah (USA) and monitored growth performance, herbivore damage and flower production. Here, plants of line ICE 1 showed a surprisingly severe growth phenotype with a 30% (±7%) reduction in rosette diameter (38 dpp p<0.0005, MWU) and 59% (±9%) reduction in stalk height (41 dpp, p<0.0001, MWU) compared to the controls (*Figure 4—figure supplement 1A*). As the heights of individual plants correlated with their log-flower production (R$^2$ = 0.86), the reduced growth of the ICE 1 plants resulted in a clear reduction in total flower numbers by 85% (±22%) compared to the controls (*Figure 4—figure supplement 1B*). In contrast, plants of the independent line ICE 6 indicated no growth differences to the controls when grown in the field. The growth phenotype of line ICE 1 could be reproduced in the glasshouse, but here it was less obvious and more attenuated compared to the field (*Figure 4—figure supplement 1C*). As we coincidently generated two versions of this line, which had their T-DNA insertions at different positions (*Figure 2—figure supplement 2B*, red and green arrows), we could demonstrate that this was independent from the insertion position, as well as independent of antimicrobial activity (*Figure 3—figure supplement 1C,D* red and green arrows) and was likely the result of a transformation or regeneration artifact. Therefore, we replaced line ICE

1 for all following experiments with the alternative line ICE 8, which showed no reduction in growth compared to the controls in the glasshouse, a similar strong transgene expression and similar peptide accumulations in roots and leaves as line ICE 6 (*Figure 2—figure supplement 3A–C*).

In a second field season we repeated the growth comparison using two independent lines (ICE 6 and ICE 8) together with empty vector controls. Unfortunately, a mishap during the germination procedure required a re-germination of the ICE 8 plants, which resulted in a delay of 12 days until this line could be planted into the field. To compare the obtained growth parameters of all lines, we

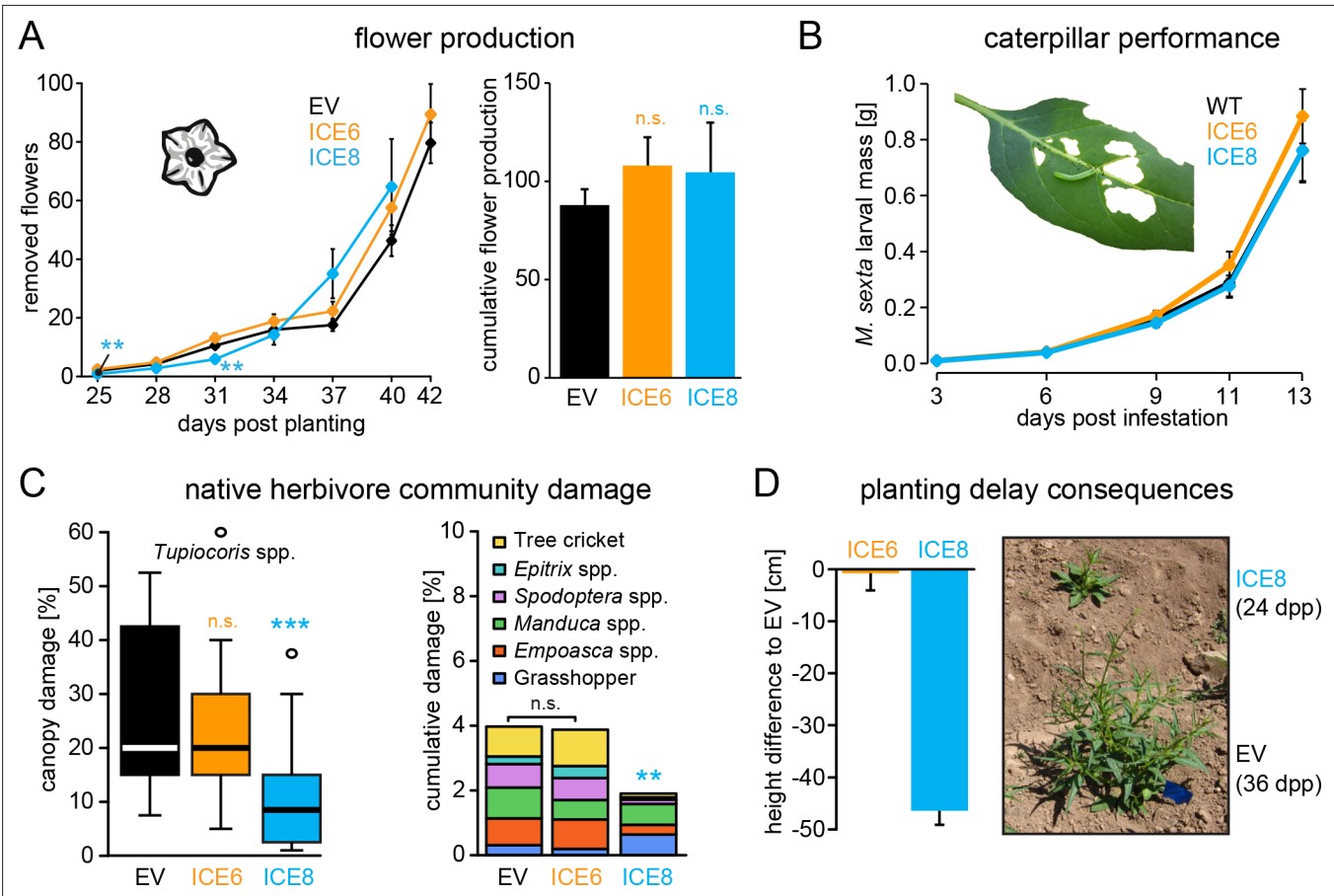

**Figure 4.** Antimicrobial peptide expression had no negative fitness consequences for field-grown plants. *N. attenuata* ICE-lines were compared with empty vector controls (EV) in a field experiment at the Lytle ranch preserve Utah (USA). (**A**) Elongated flowers and flower buds were removed before opening in a three-day interval and normalized against days post planting to compensate the planting delay of 12 days for the ICE 8 plants. For the cumulative flower production all removed flowers per genotype were summed from 25 to 40 dpp (±SEM, n = 20 plants; Mann-Whitney U Test following Kruskal-Wallis Test: **p≤0.01, n.s. = not significant). (**B**) Larval performance of the tobacco hornworm (*Manduca sexta*) did not differ in WT and transgenic ICE lines in the glasshouse (±SEM, n = 18–22 larvae). (**C**) Herbivore damage from the native herbivore community did not differ among EV and ICE 6 plants. The graphs show the estimated % canopy damage averaged from the assessments of two independent researchers (±SEM, n = 20 plants). Except for the sap-sucking herbivore *Tupiocoris* spp. (Heteroptera: *Miridae*) most herbivores showed damage with low abundance (<1%). The median is shown as the centered line, limited by the 25th and 75th percentiles and 1.5 times extended whiskers after Tukey. Only line ICE 8 (planted with delay) showed significantly less herbivore damage compared to EV (Mann-Whitney U test following Kruskal-Wallis Test; **p≤0.01; ***p≤0.001). (**D**) At the time of the herbivore screening ICE 8 and EV plants showed an average height difference of 46.5 cm (±2.7 cm) due to the delay in planting, EV and ICE 6 plants did not differ.

DOI: https://doi.org/10.7554/eLife.28715.010

The following figure supplements are available for figure 4:

**Figure supplement 1.** Off-target effects of the plant transformation process dominate growth performance of line ICE 1.

DOI: https://doi.org/10.7554/eLife.28715.011

**Figure supplement 2.** Growth performance of transgenic *N. attenuata* ICE 6 and ICE 8 lines did not differ from the controls in the field.

DOI: https://doi.org/10.7554/eLife.28715.012

normalized them against days *post* planting (dpp) (*Figure 4—figure supplement 2A*). After normalization, both ICE-genotypes indicated no differences in rosette diameter or stalk heights compared to the controls (*Figure 4—figure supplement 2B*). The average height measurements of field-grown ICE 6 and EV plants were nearly identical and showed a CV of only 2.2% (±1.4%). As an indicator for reproductive capabilities and plant fitness, we counted and removed all protruding flowers at regular intervals (*Figure 4A*). The ICE 8 plants showed only at two of the early time points a minor reduction in flower numbers (25 and 31 dpp, p<0.002, MWU), but later a similar flower production compared to the controls. To estimate the total flower production per individual plant, we summed the flower counts from 25 to 40 dpp and compared the cumulative flower production, which showed no significant differences among the genotypes (p=0.266, KW) (*Figure 4A*).

In addition to antimicrobial effects, AMPs have been demonstrated to show diverse biological functions, including the inhibition of digestive enzymes or direct insecticidal activities (*Daly and Craik, 2011*). We evaluated larval performance of the specialized herbivore (*Manduca sexta*) in the glasshouse, to exclude the possibility that the expression of the ICE peptide could have a direct effect on herbivores, as previously shown for the ectopic expression of a native plant defensin in *N. attenuata* (*Li et al., 2017*). The mass gain of these caterpillars did not differ from those feeding on the transgenic plants compared to the controls (*Figure 4B*). However, differences in the abundance of native bacterial communities have been suspected to be associated with herbivore damage (*Humphrey*

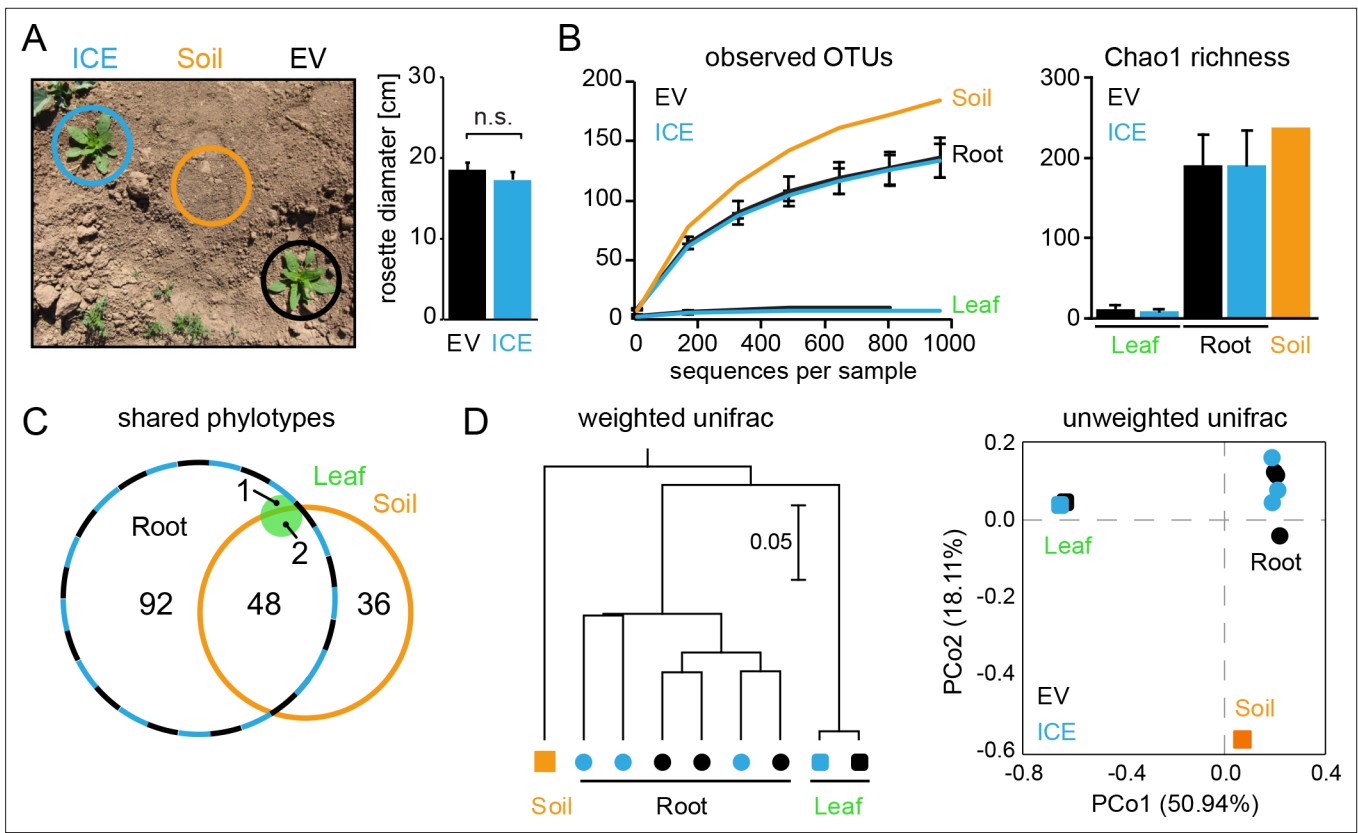

**Figure 5.** Bacterial communities from field-grown plants showed differences in species richness between the root and the leaf compartments. (**A**) Empty vector control (EV) and ICE 8 plants were planted in a paired design at the field plot on the Lytle Ranch Preserve in Utah (USA). Plant pairs showed equal growth and were harvested during the rosette stage of growth (22 dpp). Leaf and root samples were analyzed for the pilot sequencing as pooled samples from five individual plants compared to bulk soil. (**B**) Alpha diversity measures of the bacterial communities in the pilot sequencing showed an extreme low species richness within the leaf compartment (EV = black, ICE 8 = blue). For subsequent analyses, all samples were rarefied to 800 reads per sample (excluding low coverage leaf samples). (**C**) Venn diagram showing the shared phylotypes (genus level) of the rarefied communities (areas proportional). (**D**) Bacterial communities from roots were distinct from leaf and bulk soil as shown by hierarchical clustering by the Unweighted Pair Group Method with Arithmetic Mean (UPGMA) and principal coordinates analysis (PCoA) using the weighted and unweighted UniFrac as a distance measure.

DOI: https://doi.org/10.7554/eLife.28715.013

*et al., 2014*). To test, if ectopic AMP expression has any influence on the performance of the native herbivore community, we compared the leaf canopy damage of field-grown plants during the second field season. We differentiated among the frequent damage of the sap sucking mirid bugs (*Tupiocoris* spp.), and the less frequent damage of mainly chewing herbivores, including tree crickets (*Oecanthus* sp.), flea beetles (*Epitrix* spp.), noctuid larvae (*Spodoptera* spp.), hornworm larvae (*Manduca* spp.) and grasshoppers (*Acrididae*). No significant differences in herbivore damage were observed between the ICE 6 and control plants (*Figure 4C*). Only the ICE 8 plants showed a significant reduction in canopy damage compared to the controls, but on the same time a height difference of 46.5 cm (±2.7 cm) during the herbivore screen, due to the delay in planting (*Figure 4D*). As opposed to plant size, herbivore canopy damage could not be normalized against days *post* planting, and the observed differences in herbivore damage cannot be related to AMP expression but reflect the unsynchronized growth and the differences in developmental stages. At the end of the growing season, we harvested ICE 6 and EV plants from the field (whereas ICE 8 plants were left in the field to continue with the growth measurements) and compared the below and above ground biomasses. Again, we did not find significant difference among the genotypes regarding root or shoot fresh mass or the root/shoot ratio (*Figure 4—figure supplement 2C*). Likewise, we did not observe differences in plant mortality, as from the initially planted 24 individuals per group, 3–4 plants from each group had died at the end of the field season.

## Peptide expression has only marginal effects on the root-associated microbiota of field-grown plants

During a third field season, we analyze if ectopic AMP expression has an influence on the native colonization by microbial communities of field-grown plants (*Figure 5A*). To minimize the spatial variabilities within the field, plants were planted within a paired design approx. 70 cm apart and equal sized plant pairs during late rosette-stage of growth were used for analysis (*Figure 5A*). ICE 8 plants showed no growth difference to the empty vector control plants. For the field sampling we used entire roots for DNA isolation as done previously (*Santhanam et al., 2017*), which include the rhizoplane and endosphere compartment for the analysis. Bacterial community sequencing was performed by the amplification of the V5 – V8 region of the 16S rDNA for barcoded pyrosequencing by a Roche 454 System, performed by a commercial provider (MR DNA, Shallowater, TX, USA). Sequences were processed using the QIIME pipeline including quality filtering, chimera depletion and the clustering into operational taxonomic units (OTUs) using >97% sequence identity. All singletons as well as sequences assigned to organelles of the plant host were removed prior analysis. The final OTU counts were normalized by rarefaction to account for differences in sequencing depth, before the samples were compared in their community composition.

In a pilot sequencing analysis, we compared bulk soil to root and leaf compartments using pooled DNA samples. The bacterial communities of all three compartments were clearly distinguishable by their species richness and diversities (*Figure 5B*). Whereas root-associated communities showed the expected rarefaction patterns, some of the leaf samples showed surprisingly low numbers of sequencing reads and overall very poor species richness (*Figure 5B,C*). The leaf samples were dominated by a single OTU (~90% relative abundance) and basically contained only the genus *Bacillus* (relative abundances between 97.9% and 99.1% for EV and the ICE line, respectively). This dominant leaf OTU showed 100% sequence similarity to *B. tequilensis* and *B. velezensis* and 97.8% identity to *B. pumilus* DSM 1794. Still, when we included the high coverage leaf samples within the hierarchal clustering they indicated no separation by genotype. The different compartments separated well across the first principal coordinate of the PCoA plot and indicated the highest sample variability and a potential influence by genotype within the root compartment (*Figure 5D*).

To evaluate if AMP expression could have any impact on the root-associated microbial communities, we re-sequenced the non-pooled root DNA samples from 10 individual plant pairs to increase the resolution and replicate number for the main analysis (*Figure 6*). The root-samples from the two genotypes showed again highly similar diversities in the rarefaction analysis and a strong overlap of shared phylotypes (*Figure 6—figure supplement 1A–B*). At the higher taxonomic levels, the root-associated microbiota of the ICE 8 and EV plants showed overall a similar composition of relative abundances with *Actinobacteria* and *Proteobacteria* as the dominating phyla (*Figure 6B*). Sample grouping by the weighted and unweighted UniFrac distance, using principal coordinate

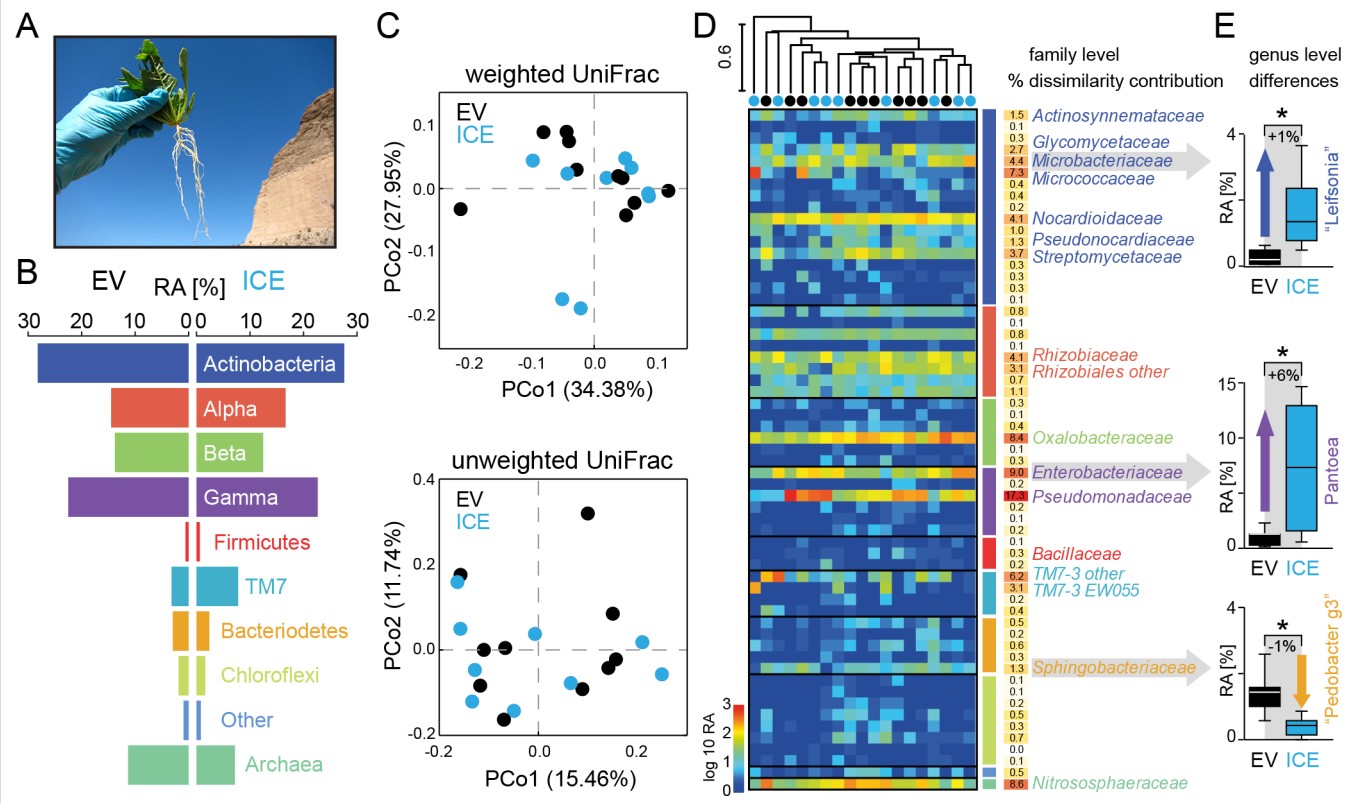

**Figure 6.** The overall composition of the root-associated microbial communities from field-grown plants did not differ between genotypes and showed marginal effects at the genus level. (**A**) Root samples from ten equal sized plant pairs of empty vector control (EV) and ICE 8 plants were used for sequencing. (**B**) Comparison of the relative abundances of the major phyla (respective classes for Protobacteria) of the root-associated microbial communities of field-grown plants. (**C**) Communities were clustered by principal coordinates analysis (PCoA) using the weighted and unweighted UniFrac as a distance measure (genotype: EV = black, ICE 8 = blue). (**D**) The filtered community data was used to visualize the distribution of all 59 families in a heatmap showing $log_{10}$ (+1) transformed abundance. Samples were clustered hierarchically by UPGMA based on Bray–Curtis dissimilarity. Similarity percentage analysis (SIMPER) was performed to evaluate sample dissimilarity contribution [%] for each family based on their relative abundance. Only families with >1% dissimilarity contribution are indicated by name. Color coding of the bars indicate higher phyla and classes as used in B. (**E**) Group significance tests revealed significant differences between control and antimicrobial peptide expressing plants at the genus level for '*Leifsonia*' (*Microbacteriaceae*), *Pantoea* (*Enterobacteriaceae*) and 'Pedobacter_g3' (*Sphingobacteraceae*) (non-parametric t-test, *p<0.05 after correcting for a false discovery rate of 10%).

DOI: https://doi.org/10.7554/eLife.28715.014

The following source data and figure supplements are available for figure 6:

**Source data 1.** Group significance comparisons from phylum to genus level.

DOI: https://doi.org/10.7554/eLife.28715.018

**Figure supplement 1.** Diversities of the root-associated microbial communities did not differ between genotypes in field-grown plants.

DOI: https://doi.org/10.7554/eLife.28715.015

**Figure supplement 2.** Diversity analysis of the root-associated microbial communities from field-grown plants separated by gram type.

DOI: https://doi.org/10.7554/eLife.28715.016

**Figure supplement 3.** Genotype differences of the root-associated communities at genus and OTU level separated by gram type.

DOI: https://doi.org/10.7554/eLife.28715.017

analysis (PCoA) revealed no separation by genotype (p=0.19, ANOSIM) (*Figure 6C*) (*Figure 6—figure supplement 1C*).

For a deeper illustration of the community composition, we filtered extremely rare OTUs from the dataset to show the relative abundances of all families within a heatmap (*Figure 6D*). Certain groups that showed overall a high abundance (e.g. *Gammaproteobacteria*) contained only a few, but dominating families (e.g. *Pseudomonadaceae* and *Enterobacteriaceae*), whereas other phyla

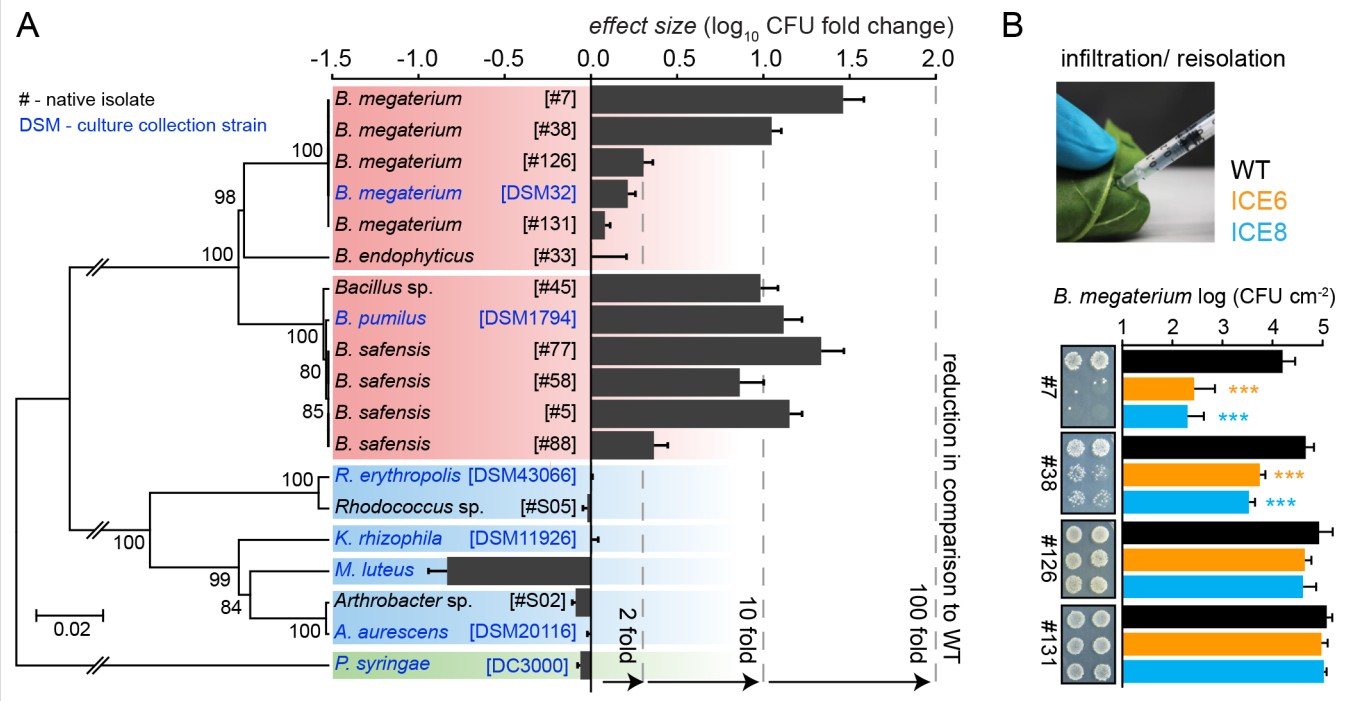

**Figure 7.** Experimental infiltrations of bacterial isolates from different taxa demonstrate the specificity of the antimicrobial activity. (**A**) Summary of the in planta antimicrobial effects obtained by individual infiltrations of bacterial isolates into transgenic and control plants, as shown in *Figure 7—figure supplement 1* and *Figure 7—figure supplement 2*. Bars represent the effect size (averaged $\log_{10}$ CFU fold reduction to WT) calculated by dividing the CFUs obtained from WT by the CFUs obtained from the transgenic plants (ICE 6 and ICE 8) from 2 to 6 dpi or 6 hpi (only *B. megaterium*). The relative phylogenetic grouping of the bacterial strains is illustrated by a neighbor-joining tree, generated from the alignment of the nearly complete 16S rDNA sequences (Figure 7—source data 1). Bootstrap supported values (1000 replicates) are indicated on individual nodes. The theoretical OTU clustering (97% similarity of the pyrosequencing amplicon) is indicated by the gaps in the background color (*Firmicutes* in red, *Actinobacteria* in blue and *Proteobacteria* in green). (**B**) Native isolates of *Bacillus megaterium* showed strain specific susceptibilities to the transgenic plants. Bacteria were injected in the leaves by pressure infiltration and re-isolated after 6 hr (±SD, n = 4 plants). Asterisks indicate statistically significant differences between WT and transgenic plants (students t-test; ***p≤0.001).

DOI: https://doi.org/10.7554/eLife.28715.019

The following source data and figure supplements are available for figure 7:

**Source data 1.** Table of bacterial isolates and culture collection strains used for experimental infiltrations.
DOI: https://doi.org/10.7554/eLife.28715.027

**Figure supplement 1.** In planta activity of *N. attenuata* ICE lines against *Actinobacteria*.

DOI: https://doi.org/10.7554/eLife.28715.020

**Figure supplement 2.** In planta activity of *N. attenuata* ICE lines against *Bacillus* isolates.

DOI: https://doi.org/10.7554/eLife.28715.021

**Figure supplement 3.** Long-term inoculations showed no difference in root colonization for *B. megaterium* nor evidence of AMP resistance development in the transgenic plants.

DOI: https://doi.org/10.7554/eLife.28715.023

(e.g. *Chloroflexi*) contained many, but low abundant families. To find the groups which accounted for major differences within the samples, we performed similarity percentage analysis (SIMPER) and obtained the sample dissimilarity contribution [%] for each family (*Figure 6D*). Mainly the high abundant families where responsible for major dissimilarity contribution, like *Pseudomona-daceae*, followed by *Enterobacteriaceae*, *Nitrososphaeraceae*, *Oxalobacteriaceae* and *Micrococcaceae*. To test if these taxa differ significantly between the genotypes, we performed individual group significance comparisons on each taxonomic level, starting from phylum to genus level, using nonparametric tests and a false discover rate of 10% (*Figure 6—source data 1*). Within the *Bacteriodetes* the family of *Sphingobacteriaceae* showed a significant reduction in the transgenic

plants compared to the controls. The same was observed at higher taxonomic ranks at the class and order levels for the *Sphingobacteriia* and *Sphingobacteriales* (*Figure 6—source data 1*). At the genus level, three different genera showed significant differences in their relative abundance between the genotypes. Interestingly, two of these showed an increase in relative abundance in the ICE line compared to the controls and were identified as '*Leifsonia*' (*Microbacteriaceae*) and *Pantoea* (*Enterobacteriaceae*) (*Figure 6E*). In particular, the relatively large increase of *Pantoea* (6% relative abundance) could largely explain the dissimilar contributions of the family *Enterobacteriaceae*. The only genus which showed a reduction in relative abundance of about 1% within the transgenic lines was found within the *Sphingobacteriaceae* and matched to an uncultured clone within the 'Pedobacter_g3' group (96% sequence similarity to *Pedobacter*). *Bacillaceae* however, showed very low relative abundances within the dataset and did not differ significantly between transgenic and control plants.

To increase the resolution of the analysis, we split the dataset by gram type and performed a separate analysis for all 'gram-positive' taxa (*Actinobacteria* and *Firmicutes*) and all 'gram-negative' taxa (containing all remaining groups). The analyses of the alpha and beta diversities showed no genotype differences when separated by gram type (ANOSIM 'gram-positive' p=0.258, 'gram-negative' p=0.091) (*Figure 6—figure supplement 2A,B*). We also used the split datasets to test for taxa which differ significantly among the genotypes. The results of the group significance comparisons at the genus level were illustrated in volcano plots showing the $\log_2$ fold difference between the genotypes (ICE vs EV) together with the $-\log_{10}$ (uncorrected) p values (*Figure 6—figure supplement 3A*). Consistent with the analysis of the entire dataset, '*Leifsonia*', 'Pedobacter_g3' and *Pantoea* differed between the genotypes at the genus level. In addition, some low abundant groups (*Dyadobacter*, *Erwinia* and *Chryseobacterium*) showed differences as well within the 'gram-negative' taxa (*Figure 6—source data 1*). When the same analysis was performed at the OTU level, only three OTUs (belonging to '*Leifsonia*' and *Patulibacter*) differed significantly within the 'gram-positive' taxa. None of the 'gram-negative' OTUs showed significant differences (*Figure 6—figure supplement 3B,C*). Interestingly, the genus *Bacillus* was the only group within the 'gram-positive' taxa which showed at least a marginal tendency for reduction from 1.3% relative abundance in the controls to 0.2% relative abundance in the transgenic plants, though not statistically significant (*Figure 6—figure supplement 3A,C*). Within the root samples, the genus *Bacillus* contained only a single OTU (OTU99) which showed 99.2% sequence identity to *B. kribbensis* and *B. deserti* and 97.4% identity to *B. megaterium*.

## Native bacterial isolates showed heterogeneous susceptibility to the transgenic plants

Since the influence of AMP expression seemed to be negligible in field-grown plants, we tested the strength of antimicrobial activity of the transgenic plants using individual native root-isolates. A plant growing in its natural habitat is exposed to a large diversity of soil bacteria and we wanted to evaluate if native isolates might show diverging levels of susceptibility, compared to cultivated laboratory strains. Altogether, we performed infiltration experiments with 12 different isolates and seven different culture collection strains. To summarize the results of all these different strains, we showed the *effect size* for each individual strain, which depicts the averaged CFU fold change to WT obtained from both independent transgenic plant lines (ICE 6 and ICE 8) (*Figure 7A*).

Since the expressed peptide was reported to be active against *Micrococcus*, we included *M. luteus* DSM 20030 and *Kocuria rhizophila* DSM 11926 as positive controls within the screening (*Tang and Gillevet, 2003*). But whereas *K. rhizophila* showed no difference to the controls, we observed for *M. luteus* at 2 dpi not a decrease but an unexpected increase in CFUs within the transgenic plants compared to the controls (*Figure 7—figure supplement 1A*). This outcome was confirmed for both transgenic plant lines in an independent infiltration experiment (*Figure 7—figure supplement 1B*). Furthermore, we tested two native *Actinobacteria* isolates (*Arthrobacter* sp. #S02 and *Rhodococcus* sp. #S05) together with closely related strains from the German culture collection. Both genera are known for their ability to degrade and metabolize nicotine, the signature defensive secondary metabolite of the plant genus (*Brandsch, 2006*; *Gong et al., 2009*), but showed no susceptibility to the peptide, even when isolated 6 days *post* inoculation (*Figure 7A*, *Figure 7—figure supplement 1C*). Overall, none of the tested *Actinobacteria* showed a reduction in the transgenic plants compared to the controls.

From the *Firmicutes* we use several *Bacillus* isolates, which were available from a previous study where they had dominated the culturable endophytic root-community of *N. attenuata* (***Long et al., 2010***). Most of the isolates belonged to the *B. pumilus*/*B. safensis* clade (strain #5, #45, #58 and #77) and showed very similar CFU reductions from 2 to 6 dpi, as the previously tested laboratory strain *B. pumilus* DSM 1794 (***Figure 7A***) (***Figure 7—figure supplement 2A***). Only isolate #88 displayed overall weaker, but still significant reductions. Consolidating all results from this clade, the antimicrobial activity of line ICE 8 tends to be stronger compared to line ICE 6 (***Figure 7—figure supplement 2A***). Experimental infiltrations with another *Bacillus* group (*B. megaterium*) indicated for the type strain DSM 32 no significant CFU reduction effects, whereas the *B. megaterium* isolate #38 revealed high levels of susceptibility (***Figure 7—figure supplement 2B***). As these differences were mainly visible at the early time points, we reduced the infiltration time for the *B. megaterium* clade and determined CFU levels at 6 hr *post* infiltration (hpi). This revealed strong susceptibilities for isolate #7 and #38, but no significant reductions for isolate #126 and #131 (***Figure 7B***). These distinct levels of susceptibility were particularly remarkable, since these isolates were nearly indistinguishable by their 16S rDNA sequences (***Figure 7A***, ***Figure 7—source data 1***). Considering that the amplicon length in pyrosequencing datasets is much shorter, and OTUs are commonly grouped by 97% sequence similarity, these strains would be grouped within a single OTU, as indicated by the subdivisions of the colored backgrounds in ***Figure 7A***.

To test if such susceptibilities would persist in a long-term colonization experiment as root-endophytes, we inoculated seeds with the two most susceptible *B. megaterium* strains (#38 and #7) and re-isolated the bacteria from surface sterilized roots after 4 weeks of growth. The obtained CFU counts of both strains showed no significant differences between the transgenic and control plants (***Figure 7—figure supplement 3A***). As we hypothesized that the bacteria could have developed AMP resistance within the roots due to the longer exposure to the peptide, we compared the 'root re-isolated' strain against the 'naïve' strain in a leaf infiltration assay. Both strains showed the same susceptibility and showed no indication for the development of peptide resistance within the transgenic plants (***Figure 7—figure supplement 3B***).

## Discussion

### Impact of AMP expression on field-grown plants – ecological vs agricultural aims

Despite the ongoing efforts in microbiome characterizations using high throughput sequencing approaches, microbiome function and its contribution to plant fitness in nature remains widely unknown. Here we report the outcome of multiple field trials where we tested if antimicrobial peptide expression for the targeted manipulation of beneficial microbes would impact plant performance in the field. Antimicrobial peptides (AMPs) are considered as promising alternatives for disease resistance engineering in crops as they show direct antimicrobial activities and are less likely to cause physiological changes within a plant compared to the expression of other resistance-genes (***de Souza Cândido et al., 2014***; ***Goyal and Mattoo, 2014***). Still, phytotoxic effects and various growth alterations have been observed in AMP expressing plants, which limits their broader application in agriculture (***Nadal et al., 2012***; ***Goyal et al., 2013***; ***Lay et al., 2014***). To be able to use transgenic plants in comparative field studies, the plants should be free of side effects that result from the plant regeneration process or from the accumulation of the peptide itself. We show that the seed specific peptide Mc-AMP1 from the common ice plant can be ectopically expressed in *N. attenuata* plants without altering morphology or growth performance of the transgenic plants. A rigorous screening revealed that the growth phenotype within one of the lines (ICE 1) was an exceptional case, as it was independent of the line´s antimicrobial activity as well as the insertion position of the transgene, and likely caused by somaclonal effects resulting from the plant transformation process. Although the growth reduction phenotype was marginal in the glasshouse, it had a large fitness impact for plants grown in the field, which illustrates not only that independently regenerated plant lines are essential for a phenotypic characterization, but also the power of field experiments for an 'augmented phenotyping' to uncover hidden traits which cannot be predicted from glasshouse experiments. A prominent example are disease-resistant wheat plants (***Zeller et al., 2010***), which showed promising yield gains in the glasshouse, but suffered from severe yield losses in the field which made these plants useless for agriculture.

We selected plant lines (ICE 6 and ICE 8) which were suitable for comparative field studies, evaluated their in planta activity against native endophytic isolates and performed in-depth phenotyping in multiple years of field trials. The advantage of an AMP expression approach (in contrast to gnotobiotic conditions) is that it allows for plant fitness estimations with respect to a plants´ ecological interactions (i.e. native herbivore community) under natural growth conditions. Despite the concerns that an unwary expression of AMPs for crop disease resistance engineering might disturb the native microbial flora, with unexpected ecological consequences for the plant, we found no evidence for growth or fitness related aberrations of AMP expression. In contrast to common expectations, AMP expression targeted against a mainly beneficial subset of bacteria seems to have negligible impact on the root-associated microbiota of field-grown plants. Agricultural interests and the depiction of AMPs as being only a part of the plant innate immune system are falling short of the various roles that AMPs play in nature, as AMPs not only function in plant protection, but also in the governance of host-symbiont interactions as shown in legume-rhizobial root nodule symbioses or even function as digestive enzyme inhibitors with direct insecticidal ability as demonstrated for a floral defensin (*Li et al., 2017*).

We used a seed-derived AMP of the knottin type family, which showed high sequence similarity to other fungal and insect peptides and to plant peptides within the order *Caryophyllales* (*Aboye et al., 2015*) (*Figure 2—figure supplement 1*). Ectopic expression of this AMP in a solanaceous species has the advantage of obviating previous adaptations of native microbial communities. Moreover, the ectopic expression also provided another dimension: AMPs have rarely been reported to be natively expressed in roots. In plants, the majority of AMPs have been isolated from seeds, followed by flowers, fruits, stems or leaves (*Carvalho and Gomes, 2009*; *Nawrot et al., 2014*; *Molesini et al., 2017*). We used a 35S promoter based system, as this is known for its high and tissue independent expression and has been preferentially used for AMP expressions in crop plants (*Holaskova et al., 2015*). Although we were able to clearly detect the heterologously expressed peptide within the root system, it showed a lower accumulation in this compartment compared to leaves. This may explain why root effects were not apparent in the transgenic plants while leaf antimicrobial effects could be readily determined.

After three field seasons, we can conclude that the ectopic expression of the antimicrobial peptide Mc-AMP1 does not influence plant growth, flower production or herbivore infestation of transgenic *N. attenuata* plants, or has any other observable negative effects on the plants when grown in the native environment. Although antibacterial effects could be demonstrated in the glasshouse in leaves, they were not apparent for roots and these experiments highlight that the in planta manipulation of complex microbial communities is a much more challenging endeavor than commonly believed.

## Divergence of the AMP activity spectrum among native isolates

In contrast to the commonly reported broad-spectrum antimicrobial effects of AMPs, we demonstrated that ectopic AMP expression in transgenic plants can show strain-specific effects and a high heterogeneity in resistance against native isolates. Newly discovered AMPs are rarely tested against a larger variety of strains or pathovars, and information about their activity spectrum is often interpolated from in vitro tests with a selected number of plant- or human-pathogens (*Aboye et al., 2015*; *Ageitos et al., 2017*). Knottins are reported to exhibit activity against gram-positive, but not gram-negative bacteria (*Pelegrini et al., 2011*), which would make them a promising candidate for the manipulation of the host-microbe relationship. However, as these groupings into activity classes were the result of in vitro tests with a few laboratory strains, it cannot be inferred that they would be able to eradicate an entire microbial group in the environment. Activity of antibiotics against laboratory strains does not commonly reflect a similar effectiveness against native bacteria (*Wright, 2007*) and for different ecosystems, the diverse communities of soil bacteria have shown to harbor non-anthropogenic resistance to various antibiotics (*Cytryn, 2013*; *Shade et al., 2013a*). As demonstrated in a previous study, AMP expressing grapevine plants showed promising resistance against a single pathogen strain in the glasshouse, but failed to display these effects in the field, which is consistent with an underestimated resilience of native pathovars (*Li et al., 2015*).

As we were interested in the effect of AMP expression against the native bacterial community, we performed experimental leaf infiltrations to evaluate the in planta antimicrobial activity of the transgenic plants using individual native isolates. The plant apoplast is an environment that is very different from that of petri dishes, and even though activity tests ex vivo had been traditionally used to confirm

correct peptide folding, they are less informative than the evaluation of whether growth inhibition could be expected *within* the plant. We could demonstrate activity against several *Bacillus* strains under physiological conditions of the leaf apoplast. But despite the strong peptide accumulations within the leaves, only about half of the tested bacterial strains showed a reduction in the experimental infiltrations (*Figure 7*). Noteworthy is *Micrococcus luteus,* which is supposed to be susceptible to this AMP from in vitro reports, but showed the opposite effect with an increase in CFU numbers within the transgenic plants. It has frequently been shown that AMP activity changes or vanishes in planta due to salt concentrations, protease-based degradation or the inhibition by phenolic compounds (*Zeitler et al., 2013*). AMPs are notoriously sensitive to divalent cations and their activity can be drastically reduced due to small amounts of $Ca^2$ or $Mg^2$ within the apoplast (*De Bolle et al., 1996*; *Güell et al., 2011*). As *Micrococcus* is usually known for its exquisite sensitivity to many antibiotics (i.e. lysozyme and insect defensins) it is commonly used as indicator strain for antimicrobial activity (*Fleming and Allison, 1922*; *Cociancich et al., 1993*). Therefore, it is not likely to predict general efficacy against other gram-positive *Actinobacteria*.

In contrast, we observed for most of the *Bacillus* isolates clear reduction effects within the leaf infiltration assay, which enabled us to use this method as a high throughput screening to select for transgenic plants with antimicrobial activity. This confirmed not only that the plants have in planta activity against *Firmicutes* but showed also a surprising heterogeneity of resistance among closely related isolates. Whereas the *B. megaterium* type strain DSM 32 showed no CFU reductions, we found strong differences in antimicrobial effects amongst native isolates (*Figure 7*). This was rather remarkable as these strains showed a high phylogenetic relatedness and were practically indistinguishable when compared by their near entire 16S rDNA sequence (*Figure 7—source data 1*). Since antimicrobial peptides are reported to target multiple essential structures of the bacterial cell, the development of resistance mechanisms is usually considered unlikely, but differences in slime capsule formation or surface charge modifications are possible explanations for the observed discrepancies among these isolates (*Nawrocki et al., 2014*). Without the experimental leaf infiltrations, such intraspecific heterogeneity would have been missed, and most culture-independent approaches would group these isolates as a single OTU. The grouping into OTUs is a technical necessarily, but it can mask a surprising diversity of ecological heterogeneity and rarely resembles a true monophyletic group (*Koeppel and Wu, 2013*). Whereas the use of single isolates is a rather simple means of accessing this diversity, it can be very rewarding as it allows to determine individual isolate functions within a population (*Savory et al., 2017*). Observing the diversity on a sub-OTU resolution via sequencing is not trivial, but recent advances point to ways of overcoming this limitation (*Tikhonov et al., 2015*; *Agler, 2016*).

When we inoculated these highly susceptible *B. megaterium* strains using a more natural inoculation assay, we did not observe reduction effects in the roots of the transgenic plants compared to the controls. This assay delivered more variable results, likely reflecting its requirement for natural colonization rather than the force-infiltration of bacteria. Although the lower peptide amount in the roots may contribute to the lack of observable direct reduction effects, we tested if it might allow bacteria to develop AMP resistance within the transgenic plants. An exposure to a low AMP concentration for 4 weeks has been shown to be sufficient to select for strains with higher AMP resistance in other studies (*Dobson et al., 2014*). However, the 'root re-isolated' strain did not differ in susceptibility compared to the original 'naïve' strain, as compared in a leaf infiltration assay, and we found no evidence consistent with the hypothesis that bacteria might develop resistances when they reside as a root endophyte within an AMP expressing plant (*Figure 7—figure supplement 3*). Clearly the results of single strain inoculations are likely poor predictors of the potential effects of AMP expression on the composition of the native community within the field.

## AMP expression has negligible impact on the native root-associated communities

The root-associated microbial communities of field-grown *N. attenuata* plants were mainly dominated by the phyla *Actinobacteria* and *Proteobacteria,* which together account for 80% of the obtained sequences (*Figure 6B*), a commonly observed pattern from the root communities of other plant species (*Bulgarelli et al., 2013*). Previous studies in Arabidopsis and maize have shown that the composition of the endophytic microbial community of these plants is mainly influenced by soil type and geographic location, whereas the plant genotype or ecotype had only a minor influence

(*Bulgarelli et al., 2012*; *Lundberg et al., 2012*; *Peiffer et al., 2013*; *Schlaeppi et al., 2014*). Even the alteration of a plant´s hormone homeostasis rarely showed any 'genotype effect' on the endophytic microbial community (*Santhanam et al., 2014*), with the exception of mutants with a heavily altered immune system (*Lebeis et al., 2015*). In contrast, we analyzed isogenic plants, which differed only in a single trait. Comparing these plants in the field, we observed no major differences within their root-associated communities. Unfortunately, *Firmicutes* which were the main target group for the generation of AMP expressing plants showed a rather low relative abundance (0.6%) in our dataset (*Figure 6B*). Bacilli were believed to be important mutualists of *N. attenuata*, as they dominated the culturable community with up to 63% relative abundance in a previous study (*Long et al., 2010*). *Bacilli* have been described for years as an important and large group of plant endophytes based on culturable approaches (*Hardoim et al., 2015*; *Levy et al., 2018*). Still, most recent NGS sequencing datasets are now pointing to much lower estimates and *Firmicutes* are therefore no longer considered as a major root inhabiting group (*Müller et al., 2016*). This trend can be seen in several studies, independent of the primer pair or sequencing platform used. The studies of *Wagner et al. (2016)* and *Robertson-Albertyn et al. (2017)* showed <1% relative abundance for *Firmicutes* in root and rhizosphere samples, when using the Illumina sequencing platform together with the primers 515F/806R. Other studies even reported a potential exclusion of *Firmicutes* from the roots of Arabidopsis and barley, as the levels in the endosphere compartment (<1% abundance) were lower than that of the surrounding soil (*Bulgarelli et al., 2012*, *Bulgarelli et al., 2015*; *Bodenhausen et al., 2013*; *Schlaeppi et al., 2014*). In a previous study, we analyzed the microbial communities of wild *N. attenuata* populations and observed a similar trend, where *Firmicutes* represented about 4.5% of the community in the bulk soil but only 0.5% for the root compartments of *N. attenuata* (*Santhanam et al., 2017*). Despite their low relative abundance, the genus *Bacillus* was the only group within the split dataset analysis which showed the tendency for a reduction among the gram-positive bacteria, though this effect could only be considered as statistically marginal (uncorrected p value 0.053) (*Figure 6—figure supplement 3A*).

Only a few studies have described *Firmicutes* as abundant community members in plants with relatively large numbers obtained from certain plant compartments of desert succulents (e.g. stem and leaf endosphere of Agavae plants) (*Coleman-Derr et al., 2016*; *Fonseca-García et al., 2016*). These leaf endosphere samples delivered the lowest sequence reads within the study of *Coleman-Derr et al. (2016)*, similar to our results from the pilot sequencing effort where we likewise sampled leaves from an arid environment. The low complexity within these samples made them less suitable for a meaningful analysis of diversity.

A few studies had investigated if transgenic plants expressing antimicrobial agents would influence plant-beneficial microbial communities, but never found clear genotype-effects (*Meyer et al., 2013*; *Turner et al., 2013*; *Kaur et al., 2017*). The feasibility of microbiome manipulation within native ecosystems appears to be more challenging than previously assumed, and even the massive application of antibiotics, as commonly done in apple orchards, had surprisingly little effect on the composition of the microbial community in soil (*Shade et al., 2013a*). Even the direct treatment of apple flowers with streptomycin, resulted in no observable change in the microbiome compositions (*Shade et al., 2013b*). This shows that our expectations of the potency of antibiotic treatments in the environment is frequently unfounded, and perhaps to be expected, given that soil bacteria are the major sources of the antibiotics that we use.

Although the overall community level showed no major shifts among the genotypes, more detailed examinations at a finer resolution revealed significant but marginal differences at the genus level. The genus 'Pedobacter_g3' within the *Sphingobacteriaceae* was the only group which showed within the ICE lines, a significant reduction in relative abundance compared to the controls. The presence of sphingolipids (the eponymous characteristic of this group) could be a potential susceptibility factor and a binding site for antimicrobial peptides (*Thevissen et al., 2000*). But as this group does not contain relevant human- or plant-pathogens, it has never been tested in AMP resistance experiments.

Interestingly, sequence read numbers matching the genera '*Leifsonia*' (*Microbacteriaceae*) and *Pantoea* (*Enterobacteriaceae*) were significantly increased within the transgenic plants compared to the controls. The genus *Pantoea* showed a particularly large increase in relative abundance of 6%. Whereas the reduction of mutualists would account for an increase in pathogens, both genera are known to contain plant pathogenic as well as beneficial bacteria and the resolution of the sequencing data does not allow inferences regarding function. As recently shown, the acquisition of virulence

plasmids can be sufficient for the transition from plant beneficial to phytopathogenic status (*Savory et al., 2017*). Little attention has been paid to microbe-microbe interactions and the importance of antagonistic mutualisms or antibiosis effects, and it cannot be excluded that under certain growth conditions AMP expression could result in an unintended increase of potential pathogenic bacteria. Greenhouse experiments with artificial communities showed that the exclusion of a single keystone species can be sufficient to influence the composition of the entire community (*Niu et al., 2017*). If this has consequences for plant fitness in the field remains to be evaluated.

## Conclusions

Although we have recently gained deep insights into the assemblage and composition of plant root microbiomes, the functional traits of major taxa remain largely unknown. Only a few bacterial groups are sufficiently characterized and recognized as important mutualists that benefit plants by enhancing growth or stress tolerance. New and innovative approaches are needed to unravel microbiome function under native growth conditions to be able to harness this potential for agricultural applications. Although ectopic AMP expression appears to be a promising approach to manipulate only certain taxa within the native community without interfering with the plant immune system, their broad-spectrum activities have rarely been tested outside the petri-dish, and expectations regarding their in planta efficacy against native communities appear overstated. This reminds us of the findings within the human gut flora, where the majority of commensals proved to be remarkably resilient to inflammation-associated AMPs (*Cullen et al., 2015*). The tremendous diversity of environmental bacteria, and their rich reservoir of genetic variability, allows natural communities to withstand antimicrobial activity and act as a single cohesive unit of antibiotic resistance as shown for marine environments (*Cordero et al., 2012*). Small mobile elements (e.g. vectors with resistance genes) can easily spread within a native microbial community, resulting in dramatic changes of resistance, with few observable shifts in the distribution of taxa. Currently, the Illumina sequencing platform has (together with the 515F/806R primer pair) become a widely used and universally accepted system for high throughput 16S rDNA profiling of bacterial communities; however, these relative short amplicons limit the phylogenetic resolution and thwarts the characterization of intraspecific variations (*van der Heijden and Schlaeppi, 2015*). The community clearly needs additional technical and computational advances in sequencing as outlined by *Agler (2016)*, and make use of longer amplicons (i.e. PacBio) (*Frank et al., 2016*), to link the 'rough' taxonomic OTU groupings of genotypes to functionally important phenotypes (*Wagner et al., 2016*). In addition, we should reconsider the utility of in vitro antimicrobial tests, as long as native bacteria could overcome such activities in vivo, not only in plants, but particularly within the human body. To paraphrase what Thomas Eisner once said about insects: Microbes won't inherit the Earth—they own it now, and it behooves us to appreciate the sophistication of their stewardship.

# Materials and methods

## Plant transformation and cultivation

For the construction of the plant transformation vector pSOL9ICE the antimicrobial peptide Mc-AMP1 of the common ice plant (*Mesembryanthemum crystallinum* L.) was selected from the PhytAMP database (http://phytamp.pfba-lab-tun.org/main.php) (ID:PHYT00272) and the cDNA sequence retrieved from NCBI (GenBank:AF069321). The gene was synthesized in sequential PCR reactions and cloned in pSOL9 binary plant transformation vectors under a constitutive cauliflower mosaic virus promoter (35S) as described in *Gase et al. (2011)* and *Weinhold et al. (2013)*. *N. attenuata* Torr. ex S. Watson seeds were originally collected in 1988 from a natural population at the DI Ranch in Southwestern Utah. Wild-type plants from the 30th inbred generation were transformed by *Agrobacterium tumefaciens*-mediated gene transfer, as described in *Krügel et al. (2002)*. For all glasshouse experiments WT seeds of the same generation were used (e.g. WT 33rd inbred generation propagated together with T3 transgenic plants), to avoid differences in seed dormancy. Seeds were surface sterilized and germinated on Gamborg's B5 Medium (Duchefa) as described in *Krügel et al. (2002)*. Seedlings were incubated in a growth chamber (Percival, day 16 hr 26°C, night 8 hr 24°C) for ten days and transferred to Teku pots, before they were transferred to 1L pots and cultivated in the glasshouse on ebb-and-flow irrigation tables under constant temperature and light conditions (day 16 hr 26–28°C, night 8 hr 22–24°C).

## Transgenic plant line screening and selection

Transgenic plant lines were screened and selected following the workflow described in *Gase et al. (2011)*, including flow cytometry, segregation analysis and diagnostic PCRs. To ensure a continuous gene expression, all plant lines affected by gene silencing were rigorously excluded as described in *Weinhold et al. (2013)*. The selected ICE lines retained full hygromycin resistance through the $T_5$ generation. If not otherwise stated, homozygous $T_4$ plants with single T-DNA insertions were used for all glasshouse experiments and plants from the $T_3$ generation for all field experiments: ICE 1 (ICE 1.1.1.1; A-09–653), ICE 6 (ICE 6.4.2.1; A-09–748) and ICE 8 (ICE 8.4.1.1; A-09–804). Ectopic expression of the transgene was confirmed for $T_4$ plants in rosette-stage leaves and in roots and shoots of seedlings by qRT-PCR. RNA was isolated by a modified salt precipitation method and qRT-PCR performed on the Mx3005P QPCR System (Stratagene) using 20 ng of cDNA from four biological replicates with the primers ICE-94F and ICE-167R as described in *Weinhold et al. (2013)*. Southern blotting was performed as previously described *Weinhold et al. (2013)* using gDNA from seedlings, digested by *Xba*I and *Eco*RV and a radiolabeled PCR fragment of the *hygromycin phosphotransferase* II gene (*hpt*II) as probe to detect the numbers of T-DNA insertions. For immunoblot analysis leaf and root samples (~100 mg) were homogenized in 300 µL extraction buffer (50 mM Tris-HCl, pH 7.5, 100 mM NaCl, 0.1% (v/v) Tween 20, 10% (v/v) glycerol, 20 mM β-mercaptoethanol and 20 µM MG132). Equal amounts (40 µg of total protein) was used for SDS-PAGE analysis (15% separation gel) and either stained by Coomassie Brilliant Blue or subjected to immunoblot analysis. The anti-ICE antibody was generated by immunizing rabbits with the peptide GCREDQGPPFCCSGF and purified by GenScript company.

## Bacterial strains

Bacterial type strains were retrieved from the German culture collection DSMZ (Deutsche Sammlung von Mikroorganismen und Zellkulturen): *Bacillus pumilus* DSM 1794, *Bacillus megaterium* DSM 32, *Kocuria rhizophila* DSM 11926, *Micrococcus luteus* DSM 20030, *Arthrobacter aurescens* DSM 20116, *Rhodococcus erythropolis* DSM 43066. Endophytic *Bacillus* spp. strains were previously isolated from *N. attenuata* plants grown in native Utah soils described in *Long et al. (2010)*. *Arthrobacter* sp. and *Rhodococcus* sp. isolates were retrieved from *N. attenuata* seedlings germinated from seeds, which had been buried for a year in a seedbank in their native habitat in Utah, USA. Surface-sterilized seeds were germinated on GB5 medium and endophytic bacteria were isolated from 15-day-old seedlings similar as described in *Long et al. (2010)*. All bacterial strains were stored at −80°C in 20% (v/v) glycerol stocks and sub-cultured on LB-Lennox agar plates (28°C). Culture media for *Pseudomonas syringae* pv *tomato* DC3000 were supplemented with the antibiotics Rifampicin (25 µg mL$^{-1}$) and Tetracyclin (5 µg mL$^{-1}$).

## Leaf infiltration and root inoculation assays

For all infiltration assays, the bacterial strains were grown as liquid overnight culture in LB-Lennox medium (28°C, 320 rpm). Bacterial cells were harvested by gentle centrifugation in a table top centrifuge (1500 × g, 2 min), the supernatant was discarded and cell pellet washed, resuspended and diluted in sterile infiltration buffer (10 mM Sodium Phosphate buffer, pH 7.0). To obtain equal CFU counts among the different taxa, overnight cultures were diluted to a final $OD_{600}$ between 0.001 and 1.0: OD 0.001 for *Pseudomonas syringae* pv *tomato* DC3000, OD 0.02 for *Bacillus pumilus*, OD 0.04 for *Arthrobacter* sp., OD 0.2 for *Rhodococcus* sp./*Kocuria rhizophila*/*Bacillus endophyticus,* OD 0.4 for *Micrococcus luteus* and OD 1.0 for *Bacillus megaterium*. Fully expanded rosette leaves (approx. 35 d old plants) were used to inject ca. 300–400 µL of bacterial solution by pressure infiltration using a 1 mL syringe without needle. Leaves were blotted dry on paper towels and the infiltrated area was marked with a pen (edding AG, Ahrensburg, Germany). For re-isolation, two leaf-discs (together 1 cm$^2$) were punched out using a cork-borer with 8 mm diameter. Most samples were taken after 0, 2, 4 and 6 d *post* infiltration (dpi) from two leaves per plant (technical replicate) and four plants per genotype (biological replicate). The 0 dpi samples were taken approximately 3 to 4 hr *post* infiltration, only the *B. megaterium* isolates were re-isolated exactly at 6 hr *post* infiltration. The leaf discs were squeezed in 400 µL sterile dilution buffer (10 mM MgCl$_2$) using a pistil in a 1.5 mL reaction tube and serially diluted to 10$^{-1}$ or 10$^{-5}$ (depending on bacterial taxon). From the three highest dilutions, 40 µL were

spotted on a square LB-Lennox agar plate and incubated at 28°C overnight (2 d for *P. syringae* and 3 d for *M. luteus*). A standard kinetic with 48 plants comprised four biological replicates × 2 technical replicates × 3 dilutions × 3 genotypes × 4 sampling days = 288 spots for CFU counting, plotted as log CFU cm$^{-2}$ leaf area. Infiltration experiments were performed multiple times and showed a similar outcome, independent from glasshouse cabin or season. The following bacterial strains had been tested for leaf infiltration, but were not used for this study, due to their slow growth or diminutive colony size: *Bacillus pichinotyi*, *Paenibacillus* sp., *Lactobacillus plantarum* and *Lysinibacillus sphaericus*.

For root inoculation, surface sterilized seeds were inoculated overnight in bacterial solution containing either *B. megaterium* isolate #38 or isolate #7 (OD$_{600}$ 1.0), diluted in 10 mM phosphate buffer (pH 7.0). Ten-day old seedlings were transferred to Tekus filled with sand. Bacteria were re-isolated at 28–30 days post inoculation (dpi) from 96 plants. Roots were surface sterilized (2 min 70% EtOH, 1 min 1–1.3% NaClO solution) and rinsed in sterile distilled water. Root samples were chopped using a flame sterilized scalpel and serial dilutions plated on LB medium. The final root wash was plated as negative control. The CFU numbers were determined after overnight incubation (n = 24 plants per group).

## Bacterial DNA isolation for 16S rDNA sequencing

Genomic DNA of bacteria was isolated using a modified CTAB method. Cell pellets from overnight cultures were pre-digested for 30 min in 450 µL Lysozyme buffer containing 20 mM Tris, 100 mM NaCl, 1 mM EDTA, 5 mg mL$^{-1}$ Lysozyme (Fluka). Cells were lysed for 30 min at 65°C in 800 µL CTAB buffer (2% CTAB, 100 mM Tris–HCl pH 8.0, 20 mM EDTA pH 8.0, 1.4 M NaCl). After addition of 500 µL chloroform, tubes were centrifuged with a tabletop centrifuge for 1 min at 16.100 g). The supernatant was again phase-separated with 700 µL chloroform after addition of 70 µL 10% CTAB solution. The aqueous phase was precipitated with 1 vol of isopropanol and the pellet washed twice in 400 µL 70% ethanol, air dried and dissolved in 50 µL nuclease-free water (Ambion). The amplification of the 16S rDNA was performed with 100 ng template DNA in a final volume of 20 µL containing 0.05 U µL$^{-1}$ JumpStart Taq DNA Polymerase using the provided reaction buffer (Sigma-Aldrich), 200 µM dNTP Mix (Fermentas) and 0.5 µM of the following primers: 27F (5'-AGAGTTTGATCCTGGCTCAG-3') and 1492R (5'-GGTTACCTTGTTACGACTT-3') (*Lane, 1991*). The amplification was performed with following program: 94°C for 1 min, followed by 29 cycles of 94°C for 30 s, 53°C for 30 s, 72°C for 30 s and a final chain elongation step of 72°C for 5 min. The PCR products were purified using the NucleoSpin Extract II kit (Macherey-Nagel) and sequenced from both sides with primers 27F and 1492R using the BigDye Terminator mix v3.1 (Applied Biosystems). The primer sequences were manually trimmed using EditSeq (DNAStar Lasergene 8) and sequences deposited in GenBank. Alignment and phylogenetic tree construction were performed with MEGA5 (*Tamura et al., 2011*) using the CLUSTALW algorithm. Bootstrap values are shown adjacent to the branches, representing the percentage support for the clusters (1000 replicates). The tree was drawn to scale, with branch lengths in the same units as those of the evolutionary distances computed using the Maximum Composite Likelihood method.

## Field experiments

Field experiments were performed on an experimental field plot at the Lytle-Ranch Preserve in Utah, USA, located within the native habitat of *N. attenuata* (Great Basin desert) (*Figure 1B*). The release of genetically modified plants was conducted in compliance with the Animal and Plant Health Inspection Service of the United States Department of Agriculture (USDA APHIS). Empty vector control plants (EV; pSOL3NC) and T$_3$ ICE-overexpression plants (ICE 6.4.2, ICE 8.4.1 and ICE 1.1.1; pSOL9ICE) were planted in three consecutive seasons under the following APHIS release numbers: 2011 (06-242-3r-a3), 2012 (11-350-101r) and 2013 (13-051-101r). The germination was carried out on Gamborg's B5 media using surface sterilized seeds. *N. attenuata* seeds showed no vertically transmitted microbes, and hence are considered to be born sterile (*Santhanam et al., 2017*). Approximately two weeks after germination, seedlings were transferred to hydrated Jiffy peat pellets and cultivated outside under half shaded conditions to adapt plants to the dry environment of the Great Basin desert. Plants were planted in the field when the root tips were protruding from the jiffies. The field plot was watered using an irrigation trench system. The entire field planting procedure can be seen in a supplemental

movie from *Santhanam et al. (2015)* (http://movie-usa.glencoesoftware.com/video/10.1073/pnas.1505765112/video-1).

## Plant growth and herbivore performance in the field

Genotypes were planted either in a randomized or paired design, as indicated for each individual experiment. As proxies of plant fitness and performance, the plant rosette diameter, stalk height and flower production were measured in 3 day intervals. The distance of the two longest leaves at opposing positions was measured as 'rosette diameter' and the length from the rosette leaves until the first flower buds measured as 'stalk height'. To avoid outcrossing of genetically modified plants, all protruding and elongated flowers were removed during counting, before the corollas could open and anthers could dehisce. For the cumulative flower production per plant, the individual flower counts within the indicated timeframe were summed up. For the correlations of plant growth with flower production the coefficient of determination ($R^2$) was determined for the log flower production to the final plant height. Experiments were terminated about 6 weeks *post* planting and flowering plants were completely excavated for the determination of the root and shoot fresh mass, whereas the ICE 8 plants were left in the field to continue with the observation of growth parameters and flower production. Damage from the native herbivore communities was estimated as percentages canopy damage per total plant leaf area and was assessed by two independent researchers. The different types of herbivores were differentiated by their characteristic damage pattern, as illustrated in *Gaquerel et al. (2013)*. Abundant herbivores were mirid bugs (*Tupiocoris* spp.), tree crickets (*Oecanthus* sp.), flea beetles (*Epitrix* spp.), noctuid larvae (*Spodoptera* spp.), hornworm larvae (*Manduca* spp.), leaf hopper (*Empoasca* spp.) and multiple genera of grasshopper (*Acrididae*). For the herbivore performance assays in the glasshouse, freshly hatched larvae of the tobacco hornworm (*Manduca sexta*) were placed on individual rosette-stage *N. attenuata* plants and caterpillar mass gain observed for 13 d of feeding (n = 18–22 larvae).

## Root sampling in the field for 16S rDNA pyrosequencing

In May 2013, we harvested plants at the rosette-stage of growth (average diameter 18.0 cm ±0.6 cm) about 3 weeks *post* planting, when the roots were well established in the soil, but before plants started elongating (average height 2.0 cm ±0.3 cm), to avoid further progression of root system lignification. As reported previously for rice, roots are colonize by bacteria within a few days of planting, and can acquire full microbiomes within about two weeks (*Edwards et al., 2015*). Bulk soil was sampled after removal of the top soil layer from about 10 cm depth from three sites within the planting area with a distance of more than 30 cm to the plants and did not contain plant roots. Equal-sized plant pairs were excavated within a block of soil to avoid tearing and loss of the fine roots. The plant roots were thoroughly washed under running tap water to remove all adhering soil particles and minimize root surface-associated bacteria, similar as done in *Wagner et al. (2016)*. Due to limitations of the sampling procedure in the field, the samples may include firmly attached rhizoplane bacteria in addition to the endosphere compartment. Three leaf samples were collected per plant and adhering dust and soil particles were removed with tap water followed by immersion in 70% isopropanol solution and finally rinsing them again with tap water. Cleaned samples were blotted on a paper towel and placed in individual paper bags for transport and preserved by desiccation as done previously (*Santhanam et al., 2017*). DNA extraction was performed using the MP-Biomedical FastDNA SPIN Kit for Soil, replacing the Fast Prep step by manual grinding of the samples in liquid nitrogen. Quantity of the DNA was evaluated on an agarose gel and by a Nanodrop spectrophotometer. Sample purity (regarding humic acid contaminations) was evaluated by the ability to amplify a 16S rDNA fragment by PCR. Samples were sent to Molecular Research LP (MR DNA, Shallowater, TX, USA) and sequenced on a Roche 454 FLX titanium instrument using the barcoded primers bac799F (5'-ACCMGGATTAGATAC-CCKG-3') and bac1394R (5'-ACGGGCGGTGTGTRC-3') as done previously (*Santhanam et al., 2014*, *Santhanam et al., 2017*). For the pilot analysis, the DNA of five plants were pooled and three pooled samples (leaves and roots) analyzed per genotype. The bulk soil sample contained DNA of three individual soil extractions. Since some of the leaf samples showed an extremely low number of reads (<40) we focused for the main analysis on the root samples and re-sequenced a higher replicate number of root samples from 10 individual plant pairs.

## Microbial community pyrosequencing data analysis

Sequence reads of root-associated microbiota were analyzed and processed using the QIIME platform (v1.8.0) (*Caporaso et al., 2010b*). Sequences were depleted of barcodes and primers and filtered for ambiguous bases and homopolymer runs exceeding 6 bp. Sequences > 200 bp were de-noised and chimeras removed using de novo and reference-based chimera detection implemented in the USEARCH function. The clustering in OTUs was performed at a 3% divergence threshold (97% similarity). Representative sequences (>200 bp) were aligned to the Greengenes 16S rDNA core set (version gg_13_5) (http://greengenes.lbl.gov) (*DeSantis et al., 2006*) using PyNAST (*Caporaso et al., 2010a*) and only sequences with a minimum identity of 70% to the closest blast hit were included in the alignment. Taxonomy was assigned using the ribosomal database project (RDP) classifier (*Wang et al., 2007*). After taxonomic assignment, all mitochondrial and chloroplast sequences were removed from the dataset by filtering for sequences matching the order 'Rickettsiales' or the phylum 'Cyanobacteria' using the filter_taxa_from_otu_table package. Likewise, all unassigned sequences (identified by manual BLAST searches as non-plastidal plant DNA) were removed from the dataset by retaining only sequences which matched to the kingdom 'Bacteria' for the pilot analysis, or 'Bacteria' and 'Archaea' for the main root analysis and resulted in a total of 24872 and 34920 quality sequences, respectively. To account for differences in sequencing depth, rarefaction analyses were conducted by randomly picking OTUs matched to the lowest OTU number found within a single sample. The pilot dataset (including soil and leaf samples) was rarefied to 800 reads. To increase the rarefaction depth for the main root-dataset, two samples with low sequence reads (ICE genotype with <600 reads) were pooled and rarefied to 724 reads. The community distances measures were calculated as weighted UniFrac based on OTU relative abundances including phylogenetic information (*Lozupone and Knight, 2005*). Non-parametric Unweighted Pair Group Method with Arithmetic Mean (UPGMA), and principal coordinate analysis (PCoA) were performed with QIIME using the weighted and unweighted UniFrac as a distance measure. We determined taxonomic differences among genotypes by Analysis of similarities (ANOSIM) tested with 99 permutations on the distance matrix and 'genotype' as category as implemented in the compare_categories package for QIIME. The rarefied root-dataset contained 333 OTUs and showed, at the genus level, 84% shared phylotypes (total of 145 genera) among the genotypes. Before group significance comparisons, an additional abundance filtering step was included to remove extremely rare taxa from the dataset using the filter_otus_from_otu_table package. Only OTUs which were either present in at least three samples, or which showed a minimum of 10 quality reads within the entire dataset were retained for the analysis. This filtering step decreased the total read number by only 3%, but removed about 40% of the low abundant OTUs. The filtered dataset was rarefied to 696 sequences and showed, at the genus level, 94% shared phylotypes (total of 101 genera) between the genotypes. The taxonomic composition (at family level) was illustrated by a heatmap using $log_{10}$ (+1) transformed relative abundances. We evaluated sample dissimilarity contribution of bacterial families based on their relative abundances, and performed similarity percentage analysis (SIMPER) with PAST v. 3.04 using the Bray-Curtis dissimilarity. To test for significant differences among the genotypes, we performed individual group significance comparisons on the filtered and rarefied dataset from phylum to genus level using non-parametric t-tests including Monte Carlo simulation with 1000 permutations as implemented in the group_significance package for QIIME. To reduce the effect on type I statistical error due to multiple comparisons of taxa we applied FDR correction using a false discovery rate ($Q$) of 10%. Similar results were obtained by analysis of variance, performed on the $log_2$ (+1) transformed dataset. For the 'split dataset' analysis, we divided the dataset by gram types and performed individual analysis for 'gram-positive' (*Actinobacteria* and *Firmicutes*) and 'gram-negative' taxa (containing all remaining groups). The datasets were filtered as described above and rarefied to 223 reads (gram-positives) and 468 reads (gram-negative), respectively. The split datasets were analyzed by group significance comparisons on genus and OTU level. Results were shown as a volcano plot using the $-log_{10}$ (uncorrected) p value and the $log_2$ fold difference calculated by dividing the relative abundances (+1 pseudocount) of the ICE lines divided by EV. Bubble size indicates relative abundance averaged from both genotypes.

## Statistical analysis

Statistical analysis was performed using Sigma Plot (v 12.0) and PAST (v 3.04) (*Hammer et al., 2001*). CFU counts were log transformed and analyzed by Student's t-tests. Plant growth parameters were

normalized against days *post* planting and compared by Kruskal–Wallis Tests (KW), followed by individual Mann-Whitney U Tests (MWU). Community sequencing data analysis was performed using the QIIME platform (v1.8.0).

## Nucleotide sequence accession numbers

Obtained sequences were deposited in GenBank under the accession numbers: KJ476709 - KJ476726 and in the European Nucleotide Archive database (accession no. PRJEB11866).

## Acknowledgement

We thank K Gase, S Kutschbach, A Wissgott for technical assistance, T Krügel and the glasshouse team for plant cultivations. HH Long and J Wu for kindly providing *Bacillus pumilus* DSM 1794 and *Pseudomonas syringae* pv *tomato* DC3000. For invaluable help with experiments, in particular for the root harvests during the field seasons, we would like to thank D Kessler, C Diezel, M Kallenbach, M Stanton, S Schuck, M Schuman, F Yon, VT Luu and others, as well as the Brigham Young University for the use of their Lytle Ranch Preserve. We thank M Schuman for critical reading of early drafts of the manuscript and the reviewers for valuable comments to improve the manuscript. A Weinhold was supported by the International Leibniz Research School for Microbial and Biomolecular Interactions (ILRS Jena), the Max-Planck Society and the European Research Council advanced grant Clockwork-Green (No. 293926) to ITB. N Rameshkumar acknowledges the support given by Max-Planck-DST India mobility grant (2015–2018).

## Additional information

### Competing interests

Ian T Baldwin: Senior Editor at *eLife*. The other authors declare that no competing interests exist.

### Funding

| Funder | Grant reference number | Author |
|---|---|---|
| European Research Council | 293926 | Ian T Baldwin |
| Max-Planck-Gesellschaft | Open-access funding | Arne Weinhold Ian T Baldwin |
| Leibniz-Gemeinschaft | | Arne Weinhold |
| Max-Planck-Gesellschaft | Max-Planck-DST India Mobility Grant | Natarajan Rameshkumar |

The funders had no role in study design, data collection and interpretation, or the decision to submit the work for publication.

### Author contributions

Arne Weinhold, Conceptualization, Formal analysis, Validation, Investigation, Methodology, Writing – original draft, Project administration; Elham Karimi Dorcheh, Validation, Investigation, Methodology, Conducted experiments required for the revision; Ran Li, Validation, Investigation, Methodology, Conducted experiments required for the revision; Natarajan Rameshkumar, Validation, Investigation, Methodology, Conducted experiments required for the revision; Ian T Baldwin, Conceptualization, Project administration, Resources, Supervision, Funding acquisition, Writing – review and editing

### Author ORCIDs

Arne Weinhold https://orcid.org/0000-0001-9999-7450
Ian T Baldwin https://orcid.org/0000-0001-5371-2974

### Decision letter and Author response

Decision letter https://doi.org/10.7554/eLife.28715.024
Author response https://doi.org/10.7554/eLife.28715.025

## Additional files

### Supplementary files

- Transparent reporting form.

DOI: https://doi.org/10.7554/eLife.28715.022

### Major datasets

The following dataset was generated:

| Author(s) | Year | Dataset title | Dataset URL | Database, license and accessibility information |
|---|---|---|---|---|
| ArneW | 2016 | Effects of antimicrobial peptide expression on the root microbiota of field-grown Nicotiana attenuata plants | http://www.ebi.ac.uk/ena/data/view/PRJEB11866 | Publicly available at the European Nucleotide Archive (accession no. PRJEB11866) |

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
