## [Decision Letter]

Thank you for submitting your article "Antimicrobial peptide expression in a wild tobacco plant reveals the limits of host-microbe-manipulations in the field" for consideration by *eLife* . Your article has been favorably evaluated by Detlef Weigel (Senior Editor) and three reviewers, one of whom is a member of our Board of Reviewing Editors.

The reviewers have discussed the reviews with one another and the Reviewing Editor has drafted this decision to help you prepare a revised submission.

Summary:

The authors Arne Weinhold and Ian Baldwin touch in their manuscript 'Antimicrobial peptide expression in a wild tobacco plant reveals the limits of host-microbe-manipulations in the field' an important topic of feasibility in translating laboratory findings to field applications. The authors therefore describe a comprehensive study analysing the effect of an antimicrobial peptide AMP (Mc-AMP1) ectopically overexpressed in *N. attenuata*. Weinhold and Baldwin perform extensive field experiments to study effects on growth, fitness, resistance against herbivores and naturally associated root microbiota and check for peptide efficiency against selected bacterial groups in the lab. They show a clear effect on certain bacteria in the lab but with strong differences between isolates of the same species. In their field experiments, the authors can show, that AMP expression does not reduce any fitness or has any other effects on the plant. To study the effect of AMP on the microbiota in the field, the authors analyse the root bacterial community using 16S amplicon sequencing. These experiments show no significant genotype effect and therefore lead the authors to conclude that Mc-AMP1 does not change bacterial composition on a genus level under field conditions which might be due to race specific resistance to antimicrobial peptides. The used amplicon method, however, does not allow race specific resolution. This hypothesis is supported by the identification of susceptible and resistant races of the same species to Mc-AMP1 in the lab.

An essential conclusion of this paper is that laboratory / glasshouse experiments cannot be translated to the field since environmental factors and microbial diversity have a higher impact than previously anticipated.

Essential Revisions:

1) One concern raised is about the primer used to do 16S rDNA profiling that might explain the unexpected low number of amplicons for *Bacillus* sp. The authors state they used a primer pair previously published by Chelius and Triplett 2001. What they used, however, is a modified primer. The original 799F primer reads AACMGGATTAGATACCCKG, the primer used in this paper reads ACCMGG...…. A primer blast in NCBI shows significant differences for both primers e.g. concerning ACCMGG how many and which *Bacillus* sp. one would retrieve. In case this is only a typo, perfect. In case this primer was really used, re-sequencing using an appropriate primer set or testing both primer sets on a defined microbial community would be needed.

2) The authors have analysed the ectopic expression of AMP1, the in planta activity against several isolates and the growth only in the phyllosphere. There is no data about the root system. Microbial communities in the field, however, were only studied in roots due to low sequence recovery for bacteria from leaves. In case there is the primer problem as mentioned in (1) leaf samples should be re-sequenced, otherwise to make sense of the root data, it would be necessary to test if roots express or secrete the same amount of AMP1 as leaves do and to use e.g. plate assays to test for bacterial inhibition by *N. attenuata* roots expressing AMP1. To use root microbial data it further needs a test to see potential growth effects on roots.

3) Another major criticism reviewers have highlighted is a focus predominantly on explaining negative results and / or controls for experimental variation, rather than testing the hypothesis. Cutting down main figures and focusing on major results that support the hypothesis would significantly improve the manuscript. Particularly differences that have been identified should be highlighted while similarities could move to supplementary data.

---

## [Author Response]

Essential Revisions:1) One concern raised is about the primer used to do 16S rDNA profiling that might explain the unexpected low number of amplicons for Bacillus sp. The authors state they used a primer pair previously published by Chelius and Triplett 2001. What they used, however, is a modified primer. The original 799F primer reads AACMGGATTAGATACCCKG, the primer used in this paper reads…. A primer blast in NCBI shows significant differences for both primers e.g. concerning ACCMGG how many and which Bacillus sp. one would retrieve. In case this is only a typo, perfect. In case this primer was really used, re-sequencing using an appropriate primer set or testing both primer sets on a defined microbial community would be needed.

About the primer sequence, we contacted Scot E. Dowd from Molecular research, the company that performed the 454 sequencing analysis for us. There seemed to be no particular reason why they offered this slightly different version of the 799F primer in their standard assay, but it had been frequently used by us and other customers (see details below). Since the mentioned mismatch would basically affect all bacterial phyla, there is no reason to assume a bias against a particular group. A single mismatch in the 5´region has little influence on the primer efficiency, in particular since the 5´end has the much bigger barcoded tail attached as well. In contrast, any mismatch in the 3´region would have a much greater influence, which is the reason why this primer is so efficient in excluding chloroplast sequences. In our experience, in silico estimations are not very good in predicting group exclusions. Take the primer Eub338 as an example. This primer has been widely used as a qPCR primer to estimate bacterial 16S copy numbers, and it has long been considered to be specific to bacteria since it contains five mismatches which are thought to exclude most of the eukaryotic sequences according to an *in silico* analysis. But as all these 5 mismatches are within the 5´ region, the experimental reality is that this primer amplifies yeast and insect host DNA with similar efficiency.

Regarding the concern about the “low” number of amplicons for *Bacillus* . Maybe we did not word this properly within the manuscript. What we meant was a much lower number of *Bacillus* reads in the sequencing analysis, compared to the expected numbers from previous culturable analysis. In the study of Long et al. (2010) up to 63% of the root isolates from *N. attenuata* were identified as *Bacillus* sp., and similar findings have been made from the endosphere compartment of other plants e.g. *Capsicum annuum* (Marasco et al. 2012). In contrast, our sequencing approach (of field grown plants) showed <1% relative abundance for *Bacillus* . This is mainly a result of cultivation bias and seems to be a common discrepancy. When we compare our sequencing results with those of other plant-microbe studies, we do not find evidence that our dataset reveals an unusually low number of *Firmicutes* . The first sequencing analysis from our department (Santhanam et al., 2014) tested different primer pairs from which the 799F/1394R combination (as used in this study) proved to be superior in competition with other commonly used primers pairs (515F/806R or 939F/1394R). With this 799F primer version, we obtained the highest species richness and the best exclusion of plant DNA and we continued using it for other studies (Santhanam et al., 2017). In this recent work, we detected in one of the soil samples up to 13% *Firmicutes* (relative abundance). On average, we found higher numbers of *Firmicutes* in the soil samples (average of 4.5% relative abundance) and a relative depletion in the root compartments of *N. attenuata* to only 0.5% relative abundance (Santhanam et al., 2017). Hence, the results that we present here (about 0.6% relative abundance of *Firmicutes* in *N. attenuata* roots) are completely consistent with our previously published results.

These levels are also within the expected range of what has been published in other studies, which rarely mention the abundance of *Firmicutes* as they are not dominant members of the community (in cultivation independent datasets). For example Peiffer et al. (2013) tested different primer pair combination, which all showed less than <2% *Firmicutes* for maize rhizosphere soil samples (estimated fromFigure 1). In the studies of Wagner et al. (2016) and Senga et al. (2017), the Illumina sequencing platform had been used with the primers 515F/806R and their data showed <0.7% *Bacillales* in soil and root samples of *Boechera stricta* and 0.5% to 1.1% *Firmicutes* in soil and rhizosphere samples of barley varieties, respectively. Donn et al. (2015) and Kawasaki et al. (2016) used exactly the same 799F primer version and the same company as we did, and detected in the wheat rhizosphere *Firmicutes* ranging from 1-2% relative abundances (estimated fromFigure 3) and for *Brachypodium* rhizosphere an average in *Firmicutes* abundance around 1.6% (compared to up to 12% in bulk soil).

Whereas Bulgarelli et al. (2015), Schlaeppi et al.(2014) and Bodenhausen et al. (2013) had used the original 799F primer version and reported from Barley and Arabidopsis, an exclusion of *Firmicutes* from the endosphere compartment with less than <1% abundance (again, estimated from their figures). Similar results have been reported previously by Bulgarelli et al. (2012) using a slightly modified 799F primer version and <1% *Firmicutes* in the root compartment. In fact, only a few studies report about an enrichment of *Firmicutes* in the root endosphere compartment of Arabidopsis (Lundberg et al. 2012). Relative high levels of *Firmicutes* have only been detected in certain plant compartments (e.g. stem and leaf endosphere of desert succulents) (Coleman-Derr et al., 2016; Fonseca-García et al., 2016).

Finally, the study of (Maes et al., 2016) used the same primer versions and the same company as we did, and successfully characterized the bee gut flora, which is highly dominated by *Firmicutes* (ranging from 20-70% relative abundance depending on tissue type). This speaks against the assumption of a primer bias, and the exclusion of this group. Although *Bacilli* have been described as the fourth largest group of endophytes when considering culturable approaches (Hardoim et al. 2015), most recent NGS sequencing datasets are now pointing to much lower estimates and *Firmicutes* are therefore no longer considered as a major root inhabiting group (Müller et al. 2016). In summary, we have no evidence that other primer combinations or sequencing platforms would be superior in detecting *Firmicutes*, or that *N. attenuata* shows unusually low levels. A short discussion of these points has been added to the Discussion section of the manuscript.

2) The authors have analysed the ectopic expression of AMP1, the in planta activity against several isolates and the growth only in the phyllosphere. There is no data about the root system. Microbial communities in the field, however, were only studied in roots due to low sequence recovery for bacteria from leaves. In case there is the primer problem as mentioned in (1) leaf samples should be re-sequenced, otherwise to make sense of the root data, it would be necessary to test if roots express or secrete the same amount of AMP1 as leaves do and to use e.g. plate assays to test for bacterial inhibition by N. attenuata roots expressing AMP1. To use root microbial data it further needs a test to see potential growth effects on roots.

To address these concerns, we performed several additional experiments, and these efforts extended the author list by three. We determined the transcript levels in roots and leaves, demonstrated peptide presence in roots and leaves using antibody-based detection and performed root inoculation assays to complement the leaf infiltration results. We also included the analysis of the pilot-sequencing dataset which included leaf and root samples from the field.

The gene expression analysis confirmed for the root and leaf compartments a strong overexpression of the transgene. Furthermore, we were successful in obtaining ICE peptide specific antibodies and were able to perform immunoblot analysis. These could finally confirm that the peptide is indeed present in the roots (Figure 2—figure supplement 3). Compared to leaf samples as positive controls, the roots seem to accumulate lower amounts of the expressed peptide, which would explain why antimicrobial effects within the roots were less apparent. The efficient generation of antibodies with specificity against AMPs has always been described as challenging in previous studies and we are delighted that it had worked out and could show that both transgenic plant lines accumulate similar peptide amounts.

To test peptide activity in the roots under semi-natural conditions, we performed a large root inoculation trial with 96 plants in the glasshouse. We used those *B. megaterium* strains which showed the highest susceptibility and had been demonstrated to be good root colonizers. Plants were grown for about 4 weeks. Despite the much larger replicate numbers (n = 24) compared to the leaf infiltration assay (n = 4) we did not observe significant differences of CFUs obtained from surface sterilized roots (Figure 7—figure supplement 3A). Whereas the lower peptide accumulations in the root system could explain why no apparent reduction effects could be observed, we further investigated an additional hypothesis, namely if peptide amounts in roots might be sufficient to select for AMP resistance in the inoculated bacteria. The inoculation time (4 weeks) was rather long, and other studies had shown that exposure to AMPs for such a time period can be sufficient to select bacteria that are adapted to AMPs (Dobson et al., 2014). We compared the strain re-isolated from the roots of transgenic plants to the original “naïve” strain. However, both strains showed no difference in their susceptibility and had nearly identical CFU reductions in a leaf infiltration assay (Figure 7—figure supplement 3B). Hence, we could confirm the antibacterial effects of the transgenic plants by leaf infiltrations, but we found no evidence that the inoculated bacteria were reduced in the roots, nor that they evolved resistance to AMPs during their 4 weeks of residence as an endophyte.

Despite substantial efforts to establish a plate bioassay for root extracts, we could not obtain the conditions or the required concentrations to demonstrate inhibitory or antimicrobial effects in vitro. As crude protein extracts contain a mixture of cytosolic content and cations which inhibit AMP activity, rigorous desalting steps are required and commonly result in peptide losses. Peptide accumulation in the roots seemed to be too low for such assay, and although we acknowledge that it might be possible to show effects by scaling up the amount of source material used for extraction, we believe that this would be of little informative value of in planta activities. It is not uncommon that up to one kilogram of starting materials has been required to obtain enough AMP for bioassays (Cammue et al., 1992; Broekaert et al., 1992), a requirement that obviates any realistic estimation in our case.

We further included the results of the pilot-sequencing of the field samples (including leaf and soil samples). This pilot-dataset was previously not included, since some leaf samples showed extreme low sequencing depth (<40 reads). Here we included all samples with >800 reads to be able to perform meaningful rarefaction and UniFrac analysis. Interestingly, previous studies on the analysis of leaf endosphere samples in Agavae plants reported also very low read recovery from such compartment (Coleman-Derr et al. 2016). In contrast to the concern of a low recovery or bias against *Firmicutes*, our leaf samples were dominated by *Bacillus* reads which comprised more than 98% of the “community” (relative abundance) from which ~90% comes only from a single *Bacillus* OTU. This resulted in a “community” with very low complexity and species richness. Rarefaction analysis did not indicate that a deeper sequencing of these samples would result in a greater resolution or a more complex structure and illustrates why we focused on the root samples and re-sequenced them for the main analysis. Not only did the root samples deliver a higher sequencing quality and reveal a real “community” structure, they showed also the largest differences in the UniFrac analysis and the largest potential for genotype differences. We tested this hypothesis with a greater number of samples from individual plant roots in our main analysis. The goal of this study was to evaluate if ectopic AMP expression would influence non-culturable community members or even a beneficial subset of the bacterial community, and plant–beneficial and growth promoting microbes have been mainly describes as root-endosphere or rhizosphere members and to a lesser extend from the leaf endosphere.

3) Another major criticism reviewers have highlighted is a focus predominantly on explaining negative results and / or controls for experimental variation, rather than testing the hypothesis. Cutting down main figures and focusing on major results that support the hypothesis would significantly improve the manuscript. Particularly differences that have been identified should be highlighted while similarities could move to supplementary data.

We suspect that this tone crept into the manuscript as a result of responding to reviewer feedback from previous cycles of review. In the light of strong preconceptions from reviewers about expected results, the text evolved in response to the increased demands for explanations for negative results and additional controls. While we are fine that the results fully support our conclusion that ectopic expression of AMPs has no influence on the growth phenotype of the plant and no major impact on the root community, the opposite is usually expected since the frequently reported “broad spectrum activity” of AMPs comes nearly exclusively from experiments using pure peptides under artificial buffer conditions. The performance of AMPs under physiological conditions can be underwhelming, and in addition to peptide degradation by proteases even a few millimolar of divalent cations within the apoplast are sufficient to completely abolish AMP activity. It was not the goal to confirm that an overdose of AMPs would have antimicrobial activity, but rather to investigate if the ectopic expression of AMPs (as used in agriculture) is sufficient to influence the native microbial community. Even the direct application of large quantities of the antibiotic streptomycin resulted, against common expectations, in no observable effect on the flower microbiome of treated apple trees (Shade et al., 2013b), or in soil of the apple orchard (Shade et al., 2013a). Our observation is fully consistent with what has been reported in previous studies showing little influence of transgenic plants expressing antimicrobial agents on the root community (Heuer et al., 2002; Rasche et al., 2006; Meyer et al., 2013).

This reminds us of the early studies which tried to investigate if mobile phone radiation might cause cancer. In a time where most people had strong expectations that this would be the case, studies with negative results were easily dismissed due to lack of sufficient experimental proof, and research programs were locked into an unproductive cycle of increasing dosage just to demonstrate effects, cycles which made mockery of the core question being studied.

We tried to improve the overall presentation of the manuscript and rewrote or removed several passages from the Discussion which distracted from the major take-home message. We updated the reference list with more recent and relevant literature and changed or added several new figures for more clarity [Figure 4 (modified), Figure 5 (new), Figure 2—figure supplement 3 (modified), Figure 4—figure supplement 2 (modified), Figure 6—figure supplement 1 (new), Figure 6—figure supplement 2 (new), Figure 6—figure supplement 3 (new), Figure 7—figure supplement 3 (new)].